# Conserved exchange of paralog proteins during neuronal differentiation

Domenico Di Fraia[1], Mihaela Anitei[1], Marie-Therese Mackmull[2,*], Luca Parca[3,*], Laura Behrendt[1], Amparo Andres-Pons[4], Darren Gilmour[5], Manuela Helmer Citterich[3], Christoph Kaether[1], Martin Beck[6], Alessandro Ori[1]

Gene duplication enables the emergence of new functions by lowering the evolutionary pressure that is posed on the ancestral genes. Previous studies have highlighted the role of specific paralog genes during cell differentiation, for example, in chromatin remodeling complexes. It remains unexplored whether similar mechanisms extend to other biological functions and whether the regulation of paralog genes is conserved across species. Here, we analyze the expression of paralogs across human tissues, during development and neuronal differentiation in fish, rodents and humans. Whereas ~80% of paralog genes are co-regulated, a subset of paralogs shows divergent expression profiles, contributing to variability of protein complexes. We identify 78 substitutions of paralog pairs that occur during neuronal differentiation and are conserved across species. Among these, we highlight a substitution between the paralogs SEC23A and SEC23B members of the COPII complex. Altering the ratio between these two genes via RNAi-mediated knockdown is sufficient to influence neuron differentiation. We propose that remodeling of the vesicular transport system via paralog substitutions is an evolutionary conserved mechanism enabling neuronal differentiation.

## Introduction

A major evolutionary event underlying the emergence of multicellular organisms is the specialization of functions between different cell types. An important role in defining the mechanisms that have led to this diversification is placed on the emergence of specific and definite gene expression programs that characterize distinct cell types (Arendt et al, 2016; Brunet & King, 2017). Multicellular organisms are characterized by an increased genome complexity, in part driven by gene duplication events (Kaessmann, 2010; Ohno, 2013). Indeed paralog genes, namely, genes that are the product of gene duplication events, are particularly enriched in the genomes of multicellular organisms (Lynch & Conery, 2003). Even though in multicellular organisms the total paralog pool is generally larger, specific cell types express only a limited subset of paralogs, indicating the existence of mechanisms that restrict the expression of some paralogs genes in a given cell type (Padawer et al, 2012). Most paralog genes share high sequence similarities and regulation of expression (Ibn-Salem et al, 2017). However, cases of divergent expression and regulation have been reported (Makova, 2003; Soria et al, 2014; Assis & Bachtrog, 2015; Brohard-Julien et al, 2021), as exemplified by the distinct roles of Hox gene family members in modulating metazoan fronto-caudal development (Ferrier & Holland, 2001). More recently, human-specific gene duplications have been described to play a role in human brain development (Suzuki et al, 2018; Schmidt et al, 2019). Besides their modulation across cell types, an important role of paralogs is reflected by their ability to compensate for each other in maintaining the general homeostatic state of cells. Genome-wide CRISPR/Cas9-based screens have shown that paralog genes have a protective action on cell proliferation against the effect of gene loss-of-function in humans (Dandage & Landry, 2019) and cancer cell lines (De Kegel & Ryan, 2019; Thompson et al, 2021). All these observations highlight the functional impact that paralog genes have in modulating biological activity, development, and cell differentiation.

From a molecular point of view, paralogs have been shown to modulate biological processes by influencing the assembly and activity of protein complexes. We have previously shown that specific compositions of protein complexes can be identified across cell types (Ori et al, 2016), and individuals (Romanov et al, 2019), and that the exchange of paralog complex members can contribute in specific cases to this variability. It has been also shown that the alternative incorporation of paralog proteins can antagonistically

[1]Leibniz Institute on Aging–Fritz Lipmann Institute, Jena, Germany  [2] Eidgenössische Technische Hochschule (ETH) Zürich Inst. f. Molekulare Systembiologie, Zürich, Switzerland  [3]Department of Biology, University of Tor Vergata, Rome, Italy  [4]European Molecular Biology Laboratory, Heidelberg, Germany  [5]Department of Molecular Life Sciences, University of Zurich, Zürich, Switzerland  [6]Department of Molecular Sociology, Max Planck Institute of Biophysics, Frankfurt am Main, Germany

Correspondence: alessandro.ori@leibniz-fli.de
*Marie-Therese Mackmull and Luca Parca contributed equally to this work.

  

modulate the function of some protein complexes. For example, multiple specific paralog substitutions between members of the BRG1- or BRM-associated factors (BAF) chromatin remodeling complex lead to the assembly of functionally distinct complexes that can influence pluripotency and neuronal differentiation (Kaeser et al, 2008; Ho et al, 2009; Son & Crabtree, 2014). Similarly, ribosomal paralog proteins promote ribosome modularity (Shi et al, 2017) and directly affect mRNA translation specificity (Slavov et al, 2015; Genuth & Barna, 2018; Gerst, 2018). Finally, co-expression analysis of protein complex members during human keratinocyte differentiation highlighted the existence of paralog subunits that compete for the same binding site in variable complexes (Toufighi et al, 2015). These studies indicate that paralog genes can contribute to the installment of specific biological functions required, for example, for cell differentiation, by influencing the activity of specific protein complexes. It remains currently unclear whether similar mechanisms extend to other molecular networks across the proteome and to which extent the regulation of paralog expression is conserved across cell types of different species. In this study, using both newly generated and publicly available datasets, we systematically investigate how the expression of paralog genes contributes to transcriptome and proteome diversification across tissues, during development and neuronal differentiation. By integrating data from multiple organisms, we define a specific signature of paralog genes that emerges during neuronal differentiation and is conserved from fish to human.

## Results

### Co-expression of paralog genes during embryo development and across human tissues

To study the contribution of gene duplication to cell and tissue variability, we analyzed the expression profiles of paralog genes during zebrafish embryonic development and across healthy human tissues. We took advantage of two publicly available datasets describing a time-course transcriptome of zebrafish embryo development (White et al, 2017), and the steady state transcriptomes and proteomes of 29 healthy human tissues (Wang et al, 2019). We used correlation analysis of transcripts and proteins encoded by all possible paralog gene pairs to address their co-regulation during development and in fully differentiated tissues. According to Ensembl Compara (Yates et al, 2020), roughly 70% of the protein coding genes in the zebrafish and human genomes have paralogs, and similar proportions of paralogs are reflected in the datasets considered in this study (71% and 74% for zebrafish and human, respectively) (Fig 1A). During both zebrafish embryo development and across human tissues, the expression profiles of most of the paralog pairs are positively correlated ($R > 0$) (Table S1). However, a substantial proportion of paralog pairs (33% and 36% for development and tissue, respectively) appears to be co-regulated in an opposite manner ($R \leq 0$) (Fig 1B and C). We define these as divergent paralog pairs. The detection of divergently regulated paralogs in RNA-Seq data can be challenging because of the handling of reads that map to multiple paralog genes. Given how these reads were handled in the analyzed datasets, our estimates

of divergent paralogs expression could be in some cases underpowered because of the equal splitting of shared reads between paralog genes (see the Materials and Methods section). Importantly, consistent results were obtained for human tissues using proteome data, which are based on proteotypic (unique) peptides that are by definition paralog-specific. The proportion of divergent paralogs estimated from the proteomics data appeared to be even higher (48%) than the estimates obtained from transcriptome data (Fig S1A).

By calculating coefficient of variations for each protein and transcript, we also noticed that genes that possess paralogs in the genome tend to be more variably expressed during development (Fig S1B) (Wilcoxon test $P < 2.2 \times 10^{-16}$), and across differentiated tissues both at the transcriptome (Fig S1C) (Wilcoxon test $P < 2.2 \times 10^{-16}$) and proteome level (Fig S1D) (Wilcoxon test $P < 2.2 \times 10^{-16}$).

Even though alternative paralog usage extends beyond protein complexes, it is already known that substitution of paralog members can contribute to the functional specialization of large protein complexes, such as chromatin remodeling complexes and ribosomes (Slavov et al, 2015; Toufighi et al, 2015; Ori et al, 2016; Romanov et al, 2019). For this reason, we analyzed paralog expression in the context of protein complexes and observed a characteristic behaviour of paralog pairs that assemble in the same complex. Specifically, we binned paralog pairs according to their reciprocal sequence identity and found that members of protein complexes display significantly lower co-expression than the paralog pairs that are not part of the same complex. This difference was especially pronounced for paralog pairs that display high sequence identity (>50%) both in the development and human tissues dataset (Fig 1D and E and Table S1). We next investigated the contribution of paralogs to the variation in composition of macromolecular complexes during development and across tissues. We calculated the median correlation between all the possible pairs of genes belonging to the same protein complex and selected the upper and lower 25% percentiles of the resulting distribution to classify protein complexes as stable or variable, respectively (Fig S2A and Table S2). For all datasets, we observed, as expected, positive correlations between protein complex members ($P < 2.2 \times 10^{-16}$, Wilcoxon test, Fig S2B). The contribution of paralogs genes to the observed variability of protein complexes is highlighted by a positive correlation between protein complex variability and paralog content, that is, the fraction of complex members that have at least one paralog in the genome ($R = 0.40$, $P = 2.1 \times 10^{-10}$) (Fig 1F and Table S2). A similar pattern can be observed using transcriptome data from human tissues ($R = 0.23$, $P = 7.9 \times 10^{-5}$) (Fig 1G and Table S2). Consistent results were also obtained when heterodimers and complexes composed only of paralog genes were excluded from the analysis (Fig S2C). Finally, by calculating pairwise co-expression of complex members (Fig S2D), we consistently observed that complex members that possess at least one paralog tend to have a more variable expression compared with other members of the same complex (Fig S2E and Table S2).

To gain insight into which biological functions are carried out by paralog pairs that show divergent expression profiles, we performed a GO Term overrepresentation analysis. We found that, within protein complexes, the anti-correlated paralog pairs (bottom 25% of the distribution) are enriched in terms related to vesicle

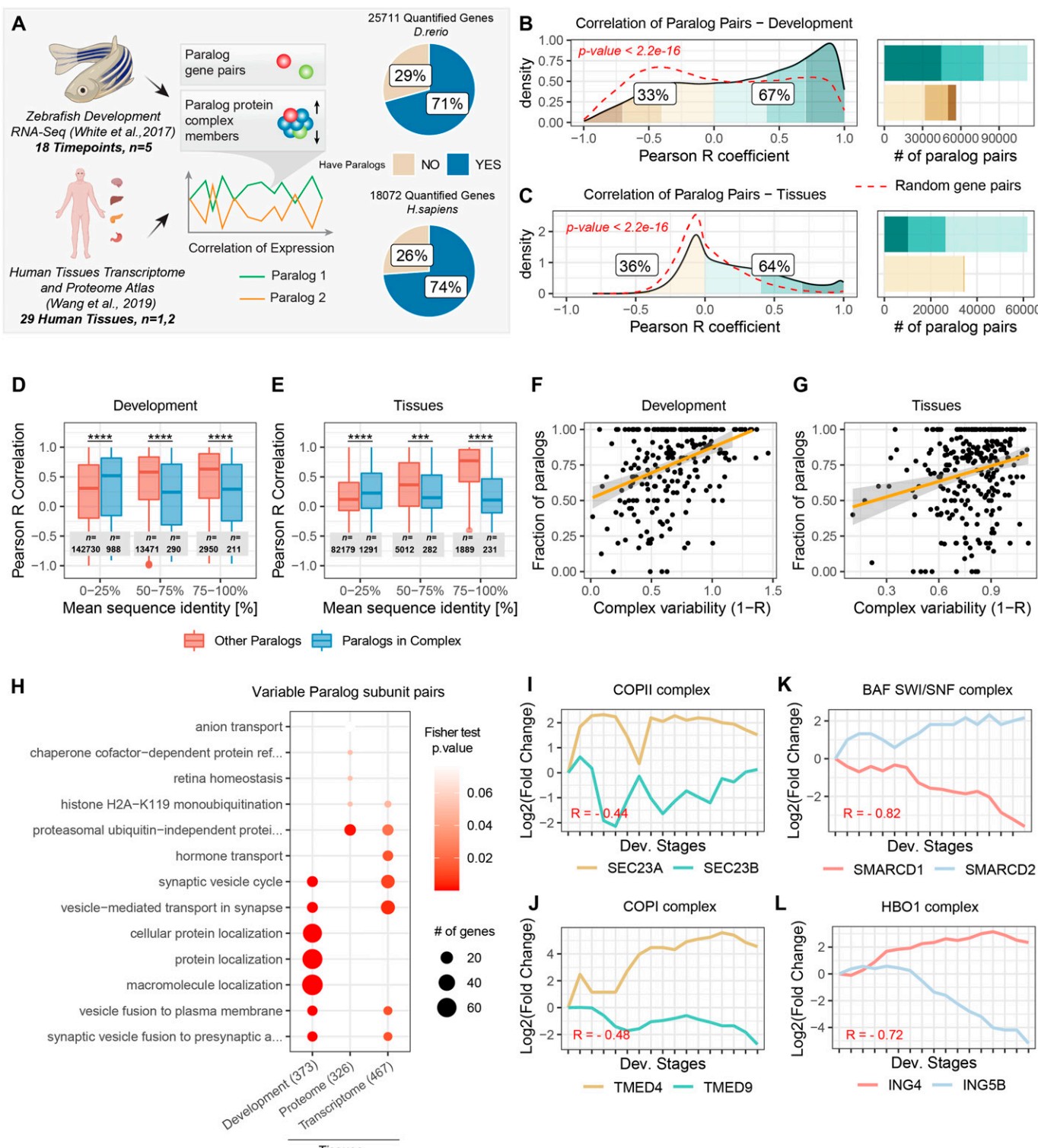

**Figure 1. Expression of paralog genes during zebrafish development and across human tissues.**
**(A)** Transcriptome data during zebrafish embryo development (White et al, 2017) and transcriptome and proteome data from 29 healthy human tissues (Wang et al, 2019) were used to calculate Pearson correlation of expression during development and across tissues for paralog gene pairs. Pie plots indicate the proportion of quantified transcripts that possess at least one paralog in the zebrafish and human dataset, respectively. **(B, C)** Density distribution of expression correlations between paralog gene pairs during zebrafish embryo development (B) and across human tissues (C). Colored areas highlight different correlation intervals. Labels indicate the percentage of paralog pairs that are positively correlated (R > 0) and negatively correlated (R ≤ 0). Barplot indicates the number of paralog pairs present in each category. Transcriptome data were used for both comparisons. Dashed red lines indicate correlation distributions of random gene pairs. *P*-values indicate the results of a two-sided Wilcoxon test between the two distributions. **(D, E)** Boxplots of expression correlation between paralog pairs displaying different levels of reciprocal sequence identity.

mediated transport (Fig 1H and Table S3). For instance, we observed divergent expression profiles for paralogs of the COPI and COPII complexes (Fig 1I and J). Our analysis also recapitulated anti-correlated expression of paralog members of the BAF chromatin remodeling complex (homologous of the yeast SWI/SNF complex [Xue et al, 2000]), which has been previously reported in other dataset (Hansson et al, 2012; Ori et al, 2016; Ho et al, 2009) (Fig 1K), but also specific expression profiles for members of the histone acetyl–transferase complex HBO1 (Fig 1L), among others (Table S1). Together, these data indicate the existence of a subset of paralog pairs, especially members of complexes involved in molecular trafficking and chromatin remodeling, that diverge in expression across developmental stages and differentiated tissues while maintaining high levels of reciprocal sequence identity.

### Transcriptional regulation and protein degradation determine the relative levels of paralog pairs

To understand which mechanisms contribute to concerted or divergent paralog regulation, we first analyzed the correlation between transcript and protein expression profiles across human tissues for individual genes. We found that genes that have at least one paralog in the genome display, on average, higher concordance between transcript and protein levels compared with genes that do not. This was true for both paralogs that are members of protein complexes (Wilcoxon test $P$-value < $2.20 \times 10^{-16}$, Fig 2A) and other paralogs (Wilcoxon test $P$-value < $2.20 \times 10^{-16}$, Fig 2A), and it indicates a substantial transcriptional control of paralog protein levels in human tissues. Consistently, co-expression profiles of paralog pairs are generally positively and significantly correlated at the transcriptome and proteome level (R = 0.39, $P$ < $2.20 \times 10^{-16}$, Fig 2B).

To assess the contribution of additional post-transcriptional mechanisms, we took advantage of a proteome kinetic analysis of protein degradation profiles (McShane et al, 2016). This work defined two major protein degradation patterns in human cells, namely proteins that are exponentially degraded and proteins that exhibit an initial rapid degradation upon synthesis followed by relatively stable levels (non-exponentially degraded). At first, we noted that paralogs that are members of protein complexes are enriched in non-exponentially degraded compared with other paralogs (Fisher test $P$-value = $4.34 \times 10^{-14}$) Fig 2C, in agreement with what was already observed for protein complex members in general (McShane et al, 2016).

Next, we assessed the relationship between protein degradation profiles and the concordance of transcript and protein co-expression for paralog pairs. To do so, we calculated a "Δ" score between the pairwise correlations of paralogs at the protein and transcript levels. Positive Δ scores indicate paralog pairs that show higher co-expression at the protein level and, conversely, negative Δ scores point to co-expression at the transcript level but divergent expression at the protein level. We found that pairs that include at least one NED paralog tend to display significantly lower "Δ" scores as compared with pairs where both proteins are exponentially degraded (Wilcoxon test $P$-value = $3.20 \times 10^{-8}$, and $P$-value = $2.41 \times 10^{-11}$ Fig 2D), suggesting that non-exponential protein degradation might contribute to determine the relative levels of this subset of paralogs, independently of transcriptional regulation.

### Conserved exchange of paralog proteins during neuronal differentiation

To investigate in more detail how the alternative usage of paralog genes contributes to cell variability, we focused on the well characterized process of neurogenesis that has been studied across different species by genome-wide approaches using both in vivo and in vitro model systems. We analyzed neurogenesis datasets from zebrafish, mouse, rat, and human (Fig 3A and Table S4), based on the hypothesis that if some particular paralog substitutions are conserved across multiple organisms, they are more likely to contribute to neuronal differentiation. We used proteomics data to account for both transcriptional and post transcriptional mechanisms regulating paralog abundances.

We generated a proteomic dataset using mouse primary neurons harvested after 0, 3 and 10 d of in vitro differentiation (DIV0, DIV3, and DIV10). Shortly, cortical immature neurons were isolated from wild-type embryonic (E15.5) mouse brains and differentiated in glia-conditioned neurobasal medium. Neurons were collected at different time points and analyzed by quantitative mass spectrometry (see the Materials and Methods section for details). We integrated this dataset with comparable data obtained from rat and human (Djuric et al, 2017; Frese et al, 2017). The rat dataset consisted of a time-course analysis of in vitro neurogenesis similar to the one performed in mouse, whereas the human data compared induced pluripotent stem cells (iPSCs), iPSC-derived neural progenitor cells (NPCs), and cortical neurons (Neu). Finally, to directly compare the proteomes of embryonic stem cells and in vivo differentiated neurons, we took advantage of an established zebrafish line that enables the isolation of intact neurons using a fluorescent reporter. In this fish strain, the red-fluorescent-protein dsRed is expressed under the control of a neuronal-specific tubulin promoter from Xenopus (NBT-dsRed) (Peri & Nüsslein-Volhard, 2008), allowing the selective isolation of neuronal cells by FACS. Undifferentiated cells

---

(D, E) Correlation values are based on transcriptome data from zebrafish development (D) and human tissues (E). Colors indicate paralog pairs that are members of the same protein complex (blue), and all other paralog pairs (red). Asterisks indicate $P$-values of the two-sided Wilcoxon test between the two compared groups: *$P$ ≤ 0.05; **$P$ ≤ 0.01, ***$P$ ≤ 0.001, ****$P$ ≤ 0.0001. (F, G) Relationship between paralogs content (fraction of complex members that have paralogs in the genome) and complex variability. Complex variability is expressed as 1-R, where R is the median Pearson correlation of expression between all complex members. (F, G) Transcriptome data were used for both zebrafish development (F) and human tissues (G). (H) GO term overrepresentation analysis for divergent paralog pairs that are members of protein complexes against all other paralog complex members. The top 5 most enriched terms from each dataset are shown. Numbers in parentheses on the x-axis indicate the number of unique divergent paralog pairs considered for enrichment. (I, J, K, L) Transcriptome profiles along embryo development for specific paralog pairs part of chromatin organization complexes, BAF/SWI (K) and HBO1 (L), or vesicle–transport complexes, COPII (I) and COPI (J). Log$_2$ fold changes calculated from TPMs relatively to the first time point are shown.

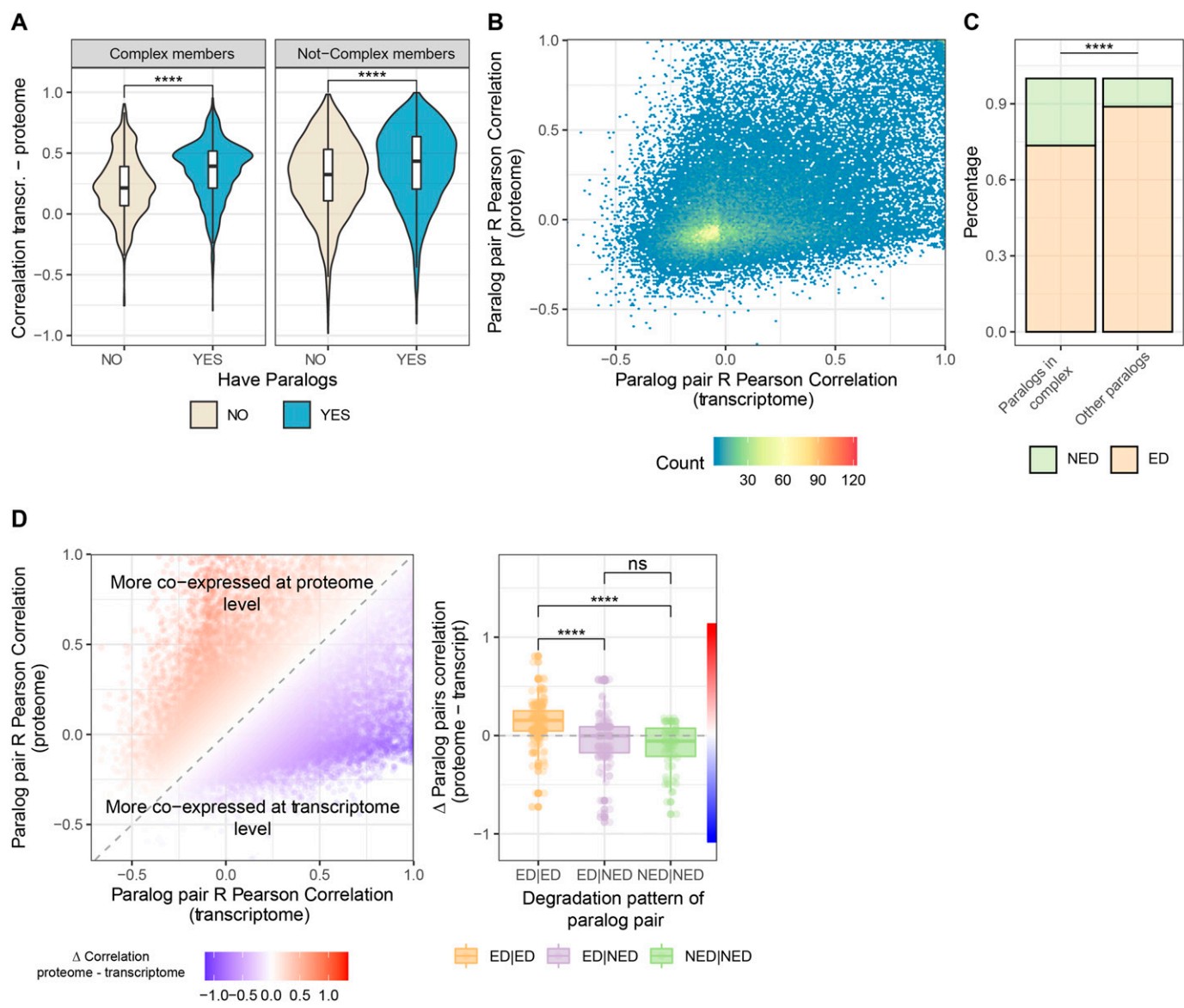

**Figure 2. Transcriptional regulation and protein degradation determine the relative levels of paralog pairs.**
**(A)** Violin plot showing distribution of Pearson correlation values between transcriptome and proteome across human tissues for genes that have paralogs (blue) and genes that do not (grey). Asterisks indicate *P*-values of the Wilcoxon test between the compared groups: ****$P$ ≤ 0.0001. **(B)** Hexbin scatterplot showing the relationship between paralog pairs co-expression (expressed as R pearson correlation between paralog pairs) at both transcriptome (x-axis) and proteome (y-axis). Color scale indicates paralog pair count in each of the represented bins of the plot. **(C)** Barplot showing the proportion of not-exponentially degraded proteins (green) and exponentially degraded proteins (orange) for proteins that have paralogs and that are either part of protein complexes or not. Asterisks indicate *P*-values of the Fisher test between the compared groups: ****$P$ ≤ 0.0001, ns, not significant. **(D)** (Left panel) Scatterplot showing the relationship between paralog pairs co-expression (expressed as R Pearson correlation between paralog pairs) at both transcriptome (x-axis) and proteome (y-axis). Color scale indicates "δ" score (differences in paralog pair correlation between transcriptome and proteome). Paralog pairs more co-expressed at transcript are represented in blue, whereas paralog pairs more coexpressed at the proteome are represented in red. Right panel: Boxplot showing the distribution of differences between proteome and transcriptome paralog pairs co-expression in relationship to their degradation profile as calculated in McShane et al (2016). Color ruler bar indicates differences in correlation between proteome and transcript as expressed in the left panel. Asterisks indicate *P*-values of the two-sided Wilcoxon test between the two compared groups: ****$P$ ≤ 0.0001, ns, not significant.

were extracted from wild-type zebrafish 6 hours post fertilization (hpf), whereas NBT-dsRed zebrafish 1 d post fertilization were used for the collection of differentiated neurons.

The quality of each dataset was evaluated using principal component analysis and GO enrichment analysis, confirming data reproducibility across replicates and the expected enrichment of terms related to neuronal development and cell differentiation (Fig

S3A–D). We validated our protein complex annotation for all organisms by showing for each dataset co-expression values against random protein complexes, obtained by randomly sampling from each proteome an equivalent amount of complexes of comparable size (Fig S4). The neuronal differentiation data recapitulated the general patterns of paralog expression that we observed during development and across tissues: (i) proteins that have at least one

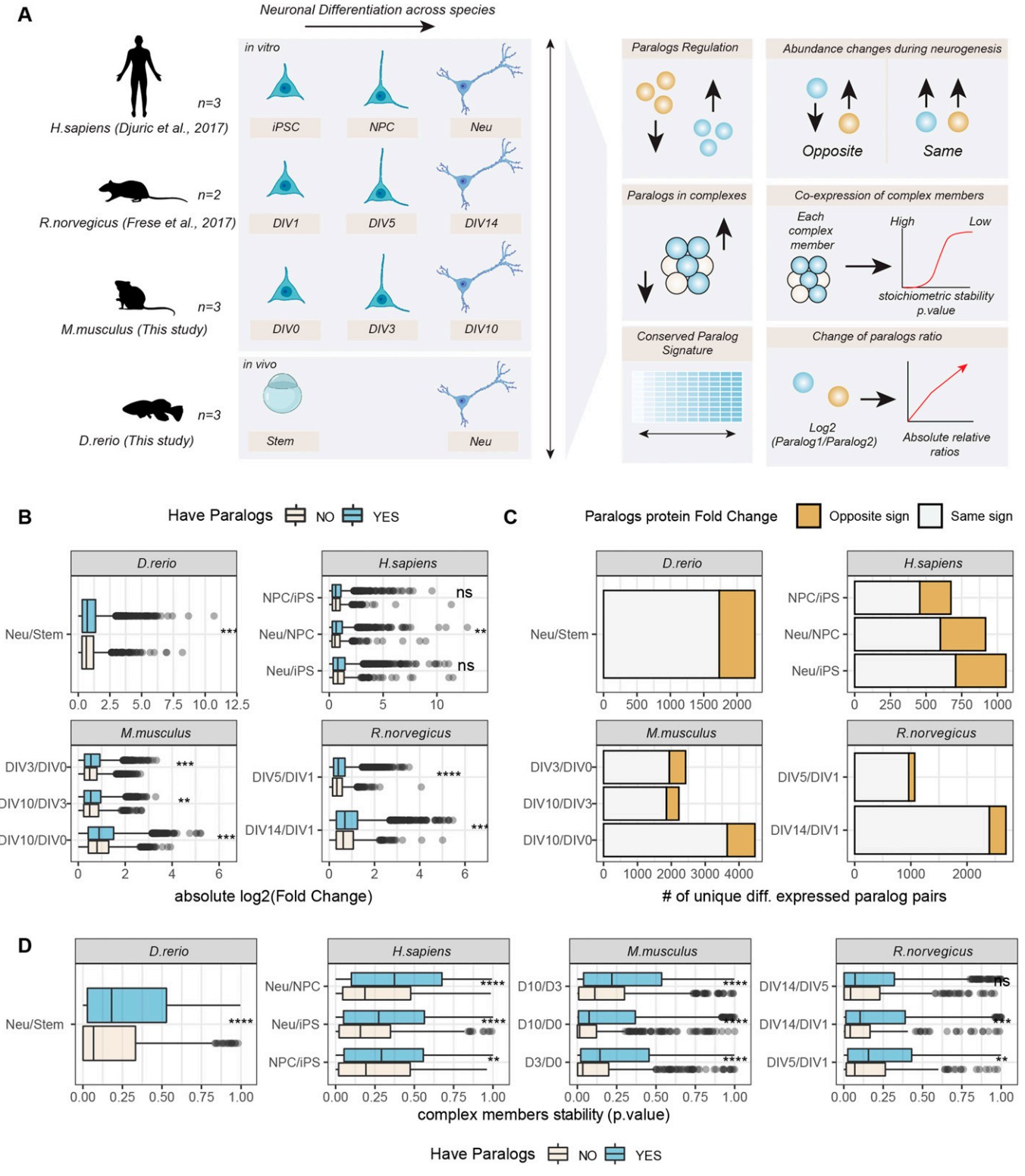

**Figure 3. Changes of abundance of paralog proteins during neuronal differentiation.**
**(A)** Overview of dataset used and data analysis workflow. DIV, differentiation in vitro day; iPSC, induced pluripotent stem cell; Neu, Neurons; NPC, neuronal precursor cell; Stem, undifferentiated stem cell. **(B)** Boxplots display absolute Log$_2$ fold changes during neuronal differentiation for proteins that have (blue) or do not have (grey) at least one paralog. **(C)** Barplots show the numbers of unique paralog pairs regulated in a concordant (grey) or opposite direction (orange) during neuronal differentiation. **(D)** Boxplots compare the stability of protein complex members that have (blue) or do not have (grey) at least one paralog in the same protein complex. Low $P$-values indicate complex members that are significantly co-expressed with the other members of the same protein complex and are therefore considered as "stable." In (B) and (D), asterisks indicate $P$-values of the two-sided Wilcoxon test between the two compared groups: *$P ≤ 0.05$; **$P ≤ 0.01$, ***$P ≤ 0.001$, ****$P ≤ 0.0001$, ns, not significant.

paralog in the genome displayed larger fold changes (Fig 3B); (ii) paralog pairs were generally co-regulated (Fig S5A and Table S5); (iii) a subset of paralog pairs (~20%) displayed opposite regulation (Fig 3C and Table S5). The latter set of paralogs was enriched for proteins related to chromatin remodeling, RNA splicing, RAS signalling, exocytosis and vesicle transport, as well as various other processes related to development. Interestingly, whereas some enrichments were dataset-specific, we consistently observed an enrichment of GO terms related to DNA binding and transport across all datasets (Fig S5B and Table S5). The neuronal differentiation datasets confirmed that paralogs contribute to protein complex variability because in general, proteins that have at least one paralog display higher stoichiometric variability (Fig 3D and Table S6), and, consequently, variable complexes were enriched in proteins with at least one paralog.

We then focused our analysis on paralog pairs that displayed divergent abundance changes during the neuronal differentiation process. To capture more subtle changes, we analyzed changes in ratios between paralog pairs across conditions using absolute protein amounts estimated from mass spectrometry data. To compare paralog pairs across species, we took advantage of the eggNOG resource (Huerta-Cepas et al, 2019). Using the eggNOG pipeline, we annotated each paralog to its orthology group (eggNOG) enabling consistent comparison of paralog genes across species. We calculated abundance ratios for all the possible paralog eggNOG pairs across conditions and we assessed significant changes in these ratios using a linear model (see the Materials and Methods section for details) (Table S7). Differences in paralog ratios were sufficient to describe the general structure of the data, as highlighted by the separation of human, rodent and zebrafish dataset by principal component analysis (Fig 4A). By mapping every paralog pair to its relative eggNOG, we compared differences in paralog ratios across datasets. At first we noticed differences in the number of paralog pairs displaying significant changes of ratios in the compared datasets, with the mouse dataset showing the largest number of detected changes (Fig S6A). We speculate that these differences might be related to heterogeneity in the proteomic workflows and experimental designs used across studies leading to different proteome coverages and limited statistical power of some of the datasets. Despite this limitation, we could identify subsets of paralog ratio changes that were common to at least two species (Fig S6B). By further applying a stringent cut-off (Log$_2$ paralog ratio differences consistent in direction in all species and at least in five of the seven condition tested, and combined adjusted $P \leq 0.05$, see the Materials and Methods section), we identified 78 paralog eggNOG pairs consistently affected during neuronal differentiation across all the species tested (Fig 4B and Table S7). These conserved paralog pairs included multiple proteins involved in redox metabolism, RNA splicing, vesicles mediated trafficking and transport. Specifically, we found changes in ratios between the COPII complex members such as SEC23A and SEC23B (Fig 4C), components of the retromer complex (VSP26B and VPS26A) (Fig S7A), dynein subunits (DYNC1LI1 and DYNC1LI2) (Fig S7B), and GTPase regulators of vesicle trafficking (RAB14 and RAB8A) (Fig S7C). Taken together, these data highlight a potential role for paralogs proteins in mediating modularity of protein complexes during neuronal differentiation. Highly conserved substitutions between paralogs appear to predominantly affect paralog pairs that participate in the formation of transport complexes. This suggests that these substitutions might be required to adapt the transport system during neuronal differentiation and development in general.

## Transcriptional regulation of paralog exchange during neuronal differentiation

Next, we asked whether the observed differential regulation of paralogs could be explained by regulatory elements in their promoter regions. Using the mouse genome as a reference, we searched for transcription factor (TF) binding motifs in a 2000-bp region upstream of the transcription starting site. For each of the conserved paralog pairs, we then quantified the percentage of shared TF-binding sites. We found that overall conserved paralog pairs shared a high percentage of regulatory elements (68%); however, a subset of TF-binding motifs (32%) could be identified only in one of the two paralogs.

We then calculated for each regulatory element differences in TF enrichment score between paralog pairs (Fig 5A). Among regulatory elements that showed greater difference in enrichment score between paralog pairs (TF identified in at least 10 of the conserved paralog pairs), we found binding motifs for multiple retinoic-acid receptors (*Rara*, *Rarb*, *Rxrb*, *Rxra*, and *Rxrg*) and TFs known to be involved in neuronal differentiation (e.g., *Sox21* and *Sox1*; Fig 5B).

We observed that groups of paralog pairs shared enrichment for specific regulatory regions (Fig 5C), indicating the possibility of a common regulation of paralog pairs by the activity of specific TFs, especially for the ones related to transport and redox processes. Together, these data suggest that differential transcriptional regulation of paralog genes by specific TFs can explain, at least in part, the paralog substitutions observed during neuronal differentiation.

## Altering the ratio between SEC23A and SEC23B affects neuronal differentiation

To experimentally test this hypothesis, we focused on the COPII members *Sec23a* and *Sec23b*. These are highly homologous paralogs that share a high level of protein sequence identity (>85%). The potentially divergent functions that these two particular paralogs may have are under debate (Zhu et al, 2015; Khoriaty et al, 2018); however, they have never been studied in the context of neuronal differentiation. Using RNAi, we knocked-down either *Sec23a* or *Sec23b* in freshly isolated mouse neurons, and analyzed the respective proteome responses during in vitro neuronal differentiation (Fig 6A). First, we confirmed that RNAi significantly reduced the protein abundance of *Sec23a* relatively to a scrambled siRNA control (Fig 6B, Log$_2$ Fold Change si*Sec23a*/siCtrl = −1.42, Qvalue = 6.83 × 10$^{-14}$, Table S8) and to a lesser extend for Sec23b (Log$_2$ Fold Change si*Sec23b*/siCtrl = −0.42, Qvalue = 1.05 × 10$^{-4}$, Table S8), globally altering the proportion between SEC23A and SEC23B in the differentiating cells. Interestingly, the knock-down of *Sec23a* induced a substantial compensatory increase of SEC23B (Log$_2$ Fold Change si*Sec23a*/siCtrl = 1.06, Qvalue = 6.06 × 10$^{-10}$, Table S8), thereby maintaining the total amount of SEC23 (summed abundance of Sec23a and Sec23b) compared with siRNA control (Fig S8A).

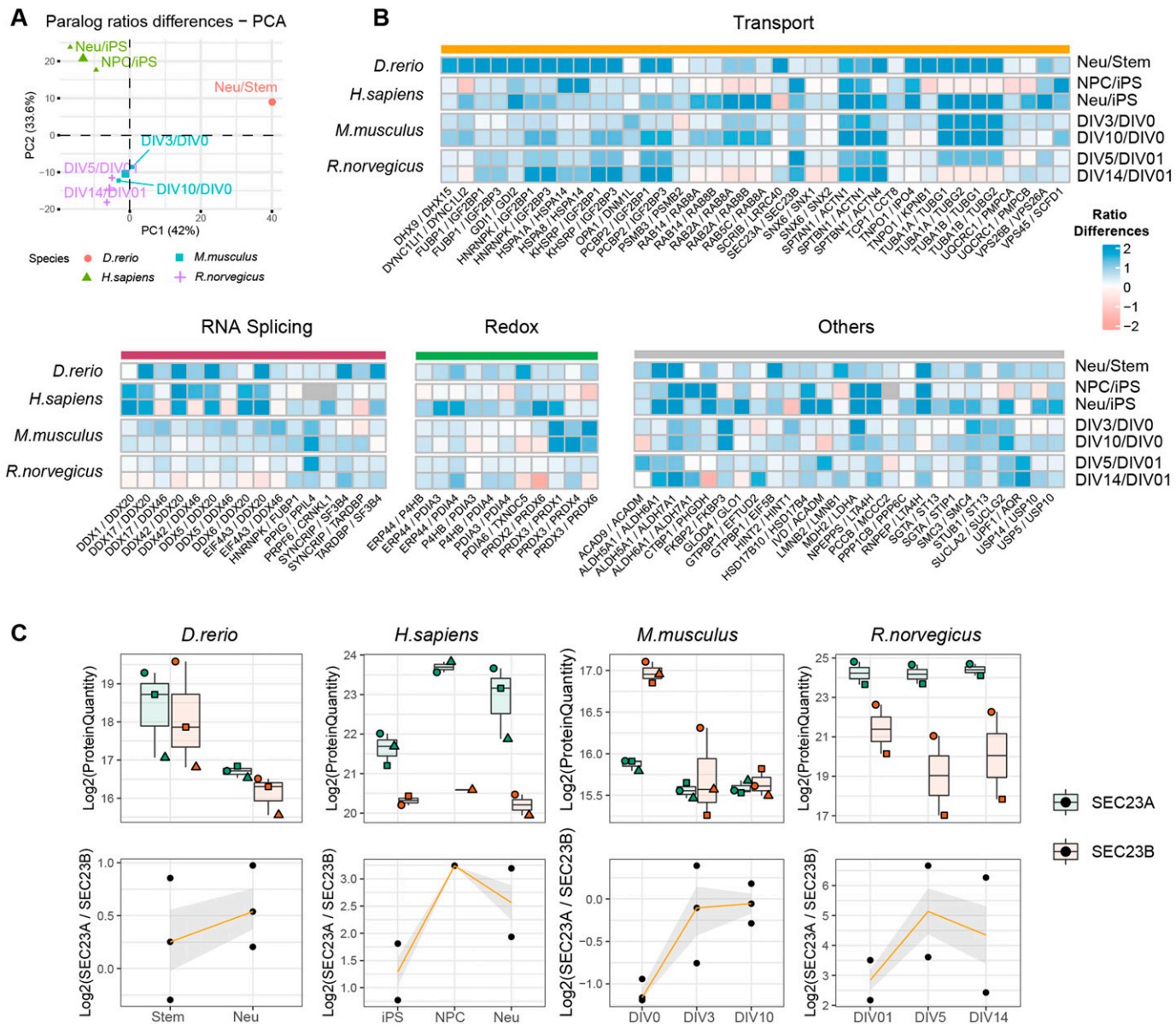

**Figure 4. A conserved paralog signature during neuronal differentiation.**
**(A)** Principal Component Analysis based on paralog ratio differences across conditions. Only paralog ratios quantified in all datasets are used for the analysis. The color code indicates the different species analyzed, the small symbols indicate the different comparisons tested, and the large symbols indicate the centroid for each species. **(B)** Heat map shows conserved paralog substitutions during neuronal differentiation. Each column represents a specific eggNOG paralog pair mapped to the same human genes. Grey tiles indicate paralog pairs not quantified in the given condition. Paralog pairs are grouped according to their known biological function. To compare ratio differences between paralogs across dataset, positive ratios were arbitrarily prioritized. **(C)** Protein abundance profiles for SEC23A (green) and SEC23B (orange) across datasets. Boxplots indicate Log$_2$ protein quantities, across different replicates, whereas line plots (bottom) indicate the ratios between the two paralogs. In the top panel, shapes indicate paired replicate experiments. In the bottom panel, orange lines indicate the mean paralog ratio across replicates, and the shaded area represents 50% confidence intervals.

A similar compensatory increase was true for the knock-down of *Sec23b*, although to a lesser extent (Fig 6B). To understand the impact of an altered balance between the SEC23 paralogs on neuronal differentiation, we compared proteome responses of the different knockdowns (KDs). The changes in protein abundance caused by the *Sec23a*-KD or *Sec23b*-KD were globally correlated when compared with siRNA control (R = 0.52, $P < 2.2 \times 10^{-16}$). However, significant paralog-specific differences could be observed

(Fig S8B). GO enrichment analysis performed on the direct comparison of *Sec23a*-KD versus *Sec23b*-KD showed that KD of *Sec23b* increased the amount of proteins closely related to neuronal activity, that is, synaptic signalling, whereas KD of *Sec23a* led to an increase in proteins related to DNA replication and RNA transcription (Fig 6C and Table S8). Among these proteins, *Sec23b* knockdown decreased the amount of BUB1B, an essential component of the mitotic checkpoint (Chan et al, 1999), as well as

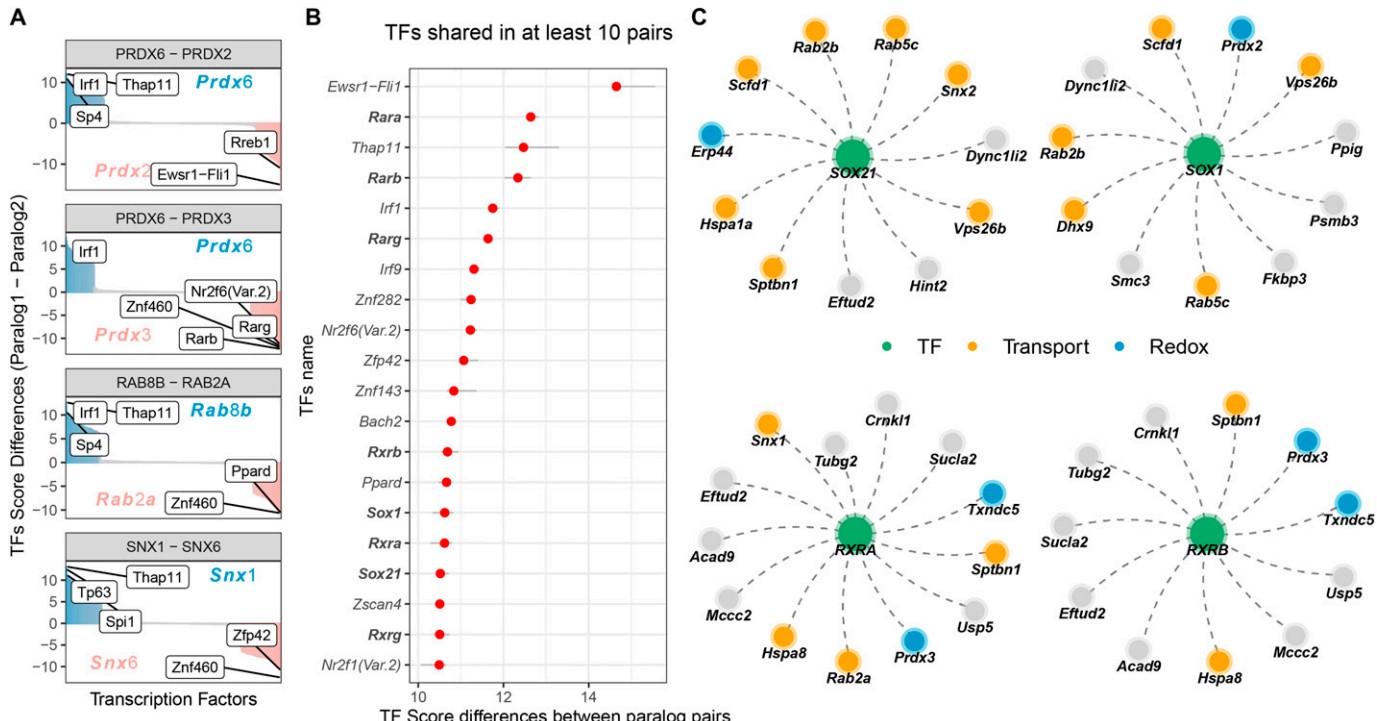

**Figure 5. Differential enrichment of transcription factor (TF)–binding sites in divergent paralog pairs.**
**(A)** Barplot showing differences in TF binding scores for selected paralog gene pairs (colored in blue and red) that display divergent expression during neuronal differentiation. Top 5 most different TFs are indicated with labels. **(B)** Distribution of differences in TF binding scores between paralog pairs. The top 20 (in terms of their median difference) TFs identified in at least 10 of the conserved paralog pairs are shown. Red dots indicate the median of the distribution. Grey lines indicate 25%–75% range of the distribution. TFs related to neuronal differentiation and retinoic acid signalling are highlighted in bold. **(C)** Network visualization for selected TFs related to neuronal differentiation and retinoic acid signalling (green) linked to paralogs for which a TF binding site was identified in their promoter region. Paralogs related to transport (orange) and redox metabolism (blue) are highlighted.

NOTCH2 a well-known regulator of cell-fate determination and known to inhibit differentiation of cerebellar neuron precursors (Solecki et al, 2001). On the other hand, it increased the levels of Synaptotagmin-1 (SYT1), a neuronal synaptic protein involved in neurotransmitter release (Coppola et al, 2001) (Fig 6D). Instead, knockdown of *Sec23a* increased the expression of the TF *Pou3f3* that has been shown to be necessary for the earliest state of neurogenesis (Sugitani, 2002; Dominguez et al, 2013) and, relatively to *Sec23b*-KD, of the component of the COP9 signalosome (MYEOV2, also known as COPS9) that has been described to promote pro-liferation (Denti et al, 2006) (Fig 6E). This pattern suggests that a higher proportion of SEC23A (as induced by the knockdown of *Sec23b*) promotes a more "neuronal" state, whereas the opposite is true for the *Sec23b* paralog, which appears to promote a more undifferentiated and proliferative state. To investigate whether these responses were more global, we directly compared the effects of *Sec23a*-KD and *Sec23b*-KD to the early changes of the proteome that occur between DIV3 and DIV0 using our mouse TMT10 data (Fig S3C). The knockdown of *Sec23a* increased the levels of proteins that are down-regulated during neuronal differentiation (KS test $P$ = 3.5 × $10^{-10}$, Fig 6F and Table S8). In contrast, the knockdown of Sec23b promoted an increase in proteins up-regulated during neuronal differentiation (KS test $P$-value = 7.1 × $10^{-5}$, Fig 6F and Table S8). This analysis confirms a functional divergence between these two paralogs, with SEC23A

promoting, and SEC23B delaying mouse neuron differentiation in vitro.

## Discussion

In this study, we characterized the specific roles that paralog genes have in promoting transcriptome and proteome variability during development, neuronal differentiation and across different tissues. In agreement with the theory that paralog genes are main carriers of biological variability (Ohno, 2013; Guschanski et al, 2017), we also found that genes that have paralogs are more often differentially expressed across tissues, during development and neuronal differentiation, indicating that they can be used as general descriptors of these specific biological states. New functional modules may then emerge in different cell types by gene duplication and subsequent functional divergence (Arendt et al, 2016; Ori et al, 2016). In agreement with this, we found that divergent expression is particularly pronounced, although not exclusive, for paralog pairs that participate in the formation of protein assemblies. More specifically, the disruption of the relationship between sequence identity and co-expression for this specific group of paralog genes could underline the existence of a specific evolutionary pressure to generate variable "modules" that adapt the function of specific protein complexes in a context-dependent manner. Consistently,

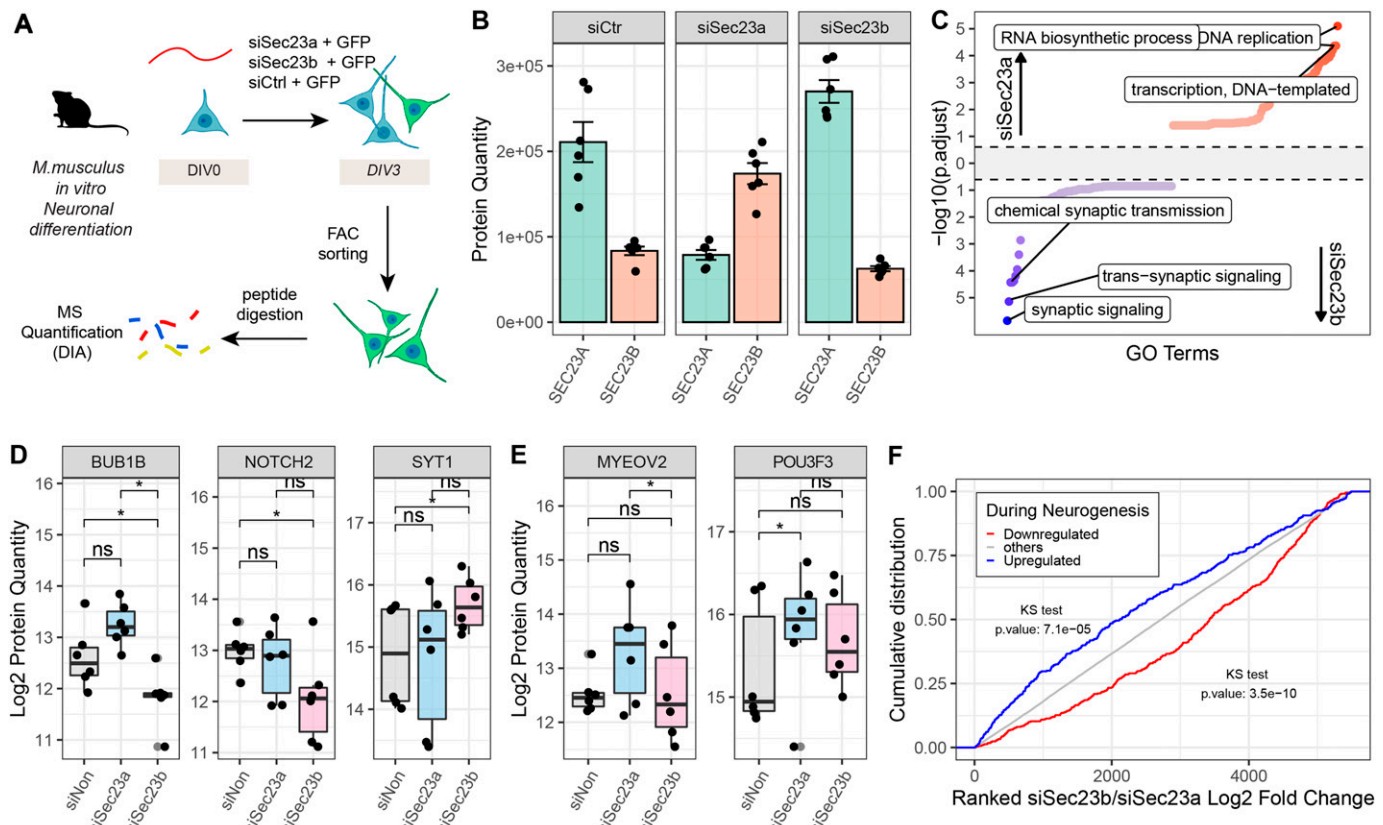

**Figure 6. Altering the ratio between SEC23A and SEC23B affects neuronal differentiation in vitro.**
**(A)** Mouse cortical neurons isolated from mouse embryos were transfected either with siCtr, siSec23a, or siSec23b, and a GFP expressing plasmid, and differentiated for 3 d. Transfected cells were isolated via FACS based on GFP expression and their proteomes analized by quantitative mass spectrometry (MS) using data-independent acquisition (DIA). **(B)** Protein abundance of Sec2a (green) and Sec23b (orange) following different siRNA treatments, estimated from mass spectrometry data. n = 6 biological replicates, from three independent isolations. **(C)** Gene Set Enrichment Analysis for "Biological Process" category of differentially abundant proteins in si*Sec23b* versus si*Sec23a*. The x-axis represents the GO terms ranked by their –log₁₀ adjusted *P*-value, for the two conditions, whereas the y-axis represents the –log₁₀(adjusted *P*-value) for each term. Top 100 GO terms enriched among proteins that are more abundant in the si*Sec23a* or si*Sec23b* condition are highlighted in red and blue, respectively. **(D, E)** Quantification of selected proteins that were differentially affected by si*Sec23b* and si*Sec23a*. Asterisk indicates *P*-values from a paired *t* test run at the precursor level and corrected from multiple testing as implemented in the Spectronaut software (see the Materials and Methods section for details). *$P \le 0.05$; **$P \le 0.01$, ***$P \le 0.001$, ****$P \le 0.0001$, ns, not significant. n = 6 biological replicates, from three independent isolations. **(F)** Cumulative distributions of ranked Log₂ fold changes (si*Sec23b*/si*Sec23a*) for proteins that are up-regulated (blue) (Log₂ FoldChange DIV3/DIV0 ≥ 1 and adjusted *P* ≤ 0.05), or down-regulated (red) (Log₂ FoldChange DIV3/DIV0 <= −1 and adjusted *P* ≤ 0.05) during mouse neuronal differentiation.

stoichiometrically variable complexes are the ones with the highest paralog content, and they are often associated with functions related to membrane trafficking and chromatin organization. The observed modularity could be then comparable to what has been described for other cellular compartments, such as vertebrate synapses, where gene duplication of scaffold synaptic proteins has been related to the emergence of complex cognitive behaviours (Nithiananantharajah et al, 2013). The divergent expression of paralog pairs that we describe implies changes of their relative abundances across developmental stages/cell types. We define such changes as paralog "substitutions" or "exchanges." However, we observed a broad range of effect sizes and, in multiple cases, both paralogs remain expressed. We believe that such substitutions reflect rather a fine-tuning than a qualitative switch of, for example, protein complex function. The analysis of human tissues for which we had both transcriptome and proteome data available indicated that transcriptional regulation has a major contribution to determine the co-expression of paralog pairs. However, we also found

evidence that post-transcriptional mechanisms, such as not-exponential protein degradation, participate in determining the relative levels of paralog proteins, as already suggested more in general for protein complexes (McShane et al, 2016).

By integrating proteomic datasets from different species, we have identified patterns of paralog regulation that occur during neuronal differentiation in multiple vertebrates. Despite heterogeneity in the cell types and developmental stages compared as well as technical differences between dataset that might have limited our ability to accurately quantify specific paralogs, we were able to extract a signature of paralog substitutions based on the detection of consistent abundance changes across species. The enrichment of paralogs involved in the transport of macromolecules supports the hypothesis of fine tuning of membrane trafficking-related functions during neuronal differentiation. The relevance of paralog divergence in trafficking complexes has been also recently highlighted by the finding that two members of the COPI complex, COPG1 and COPG2, play distinct roles in modulating

mouse neurogenesis (Goyal et al, 2020). This specific substitution was also clearly identified in our mouse data, but not in all other datasets, suggesting that some of these functions could also be species specific. Moreover, we also addressed a similar exchange between the COPII complex members SEC23A and SEC23B that was highly conserved during neuronal differentiation from fish to humans. Previous studies on the functional divergence of these two paralogs reached contradicting conclusions, depending on the model system investigated. Some studies, carried out by substituting *Sec23a* in the *Sec23b* gene locus, have proposed a complete functional overlap of these two proteins (Khoriaty et al, 2018). Works by others have indicated separate roles regarding the ability to transport receptors (Scharaw et al, 2016) and cargo substrates (Zeng et al, 2015; Zhu et al, 2015). Although these two paralogs are still highly redundant in function, we observed that they carry out different roles in respect to neuronal differentiation, with the SEC23A paralogs being needed to correctly progress during the neuronal differentiation process. Knockdown of either of the two paralogs induced opposite responses during in vitro neuronal differentiation, suggesting that a balanced paralog ratio is needed to correctly modulate this process.

More generally, our study has highlighted the importance of paralog gene pairs in neuronal differentiation, as we have illustrated the possibility of promoting or antagonizing neuronal differentiation by targeting specific paralog genes. Similar mechanisms might be valid in other cell types or in different biological states, including pathological ones. Understanding which paralog genes define different cell identities could be exploited in the future for transdifferentiation purposes, for example, for the generation of new models of neurodegenerative diseases (Mertens et al, 2018). In this case, we can speculate that specific paralog substitutions could help drive lineage transition between different somatic cells. However, broader comparisons between different cell types, integrating multiple data sources, single-cell analyses, and functional studies of specific paralogs are needed to better elucidate all these different possibilities.

# Materials and Methods

## Dataset and resources

### Ensembl Compara paralog genes resources
Paralogs annotation for *Homo sapiens* (GRch38.p13) *Danio rerio* (GRCz11) *Mus musculus* (GRCm38.p6) *Rattus norvegicus* (Rnor_6.0), were downloaded from Ensembl (v102) via biomart (http://www.ensembl.org/biomart/martview/f04b3aa8b5c7f463e3edf9fa58d205a7). Duplicated paralog pairs (e.g., Paralog$_1$ | Paralog$_2$; Paralog$_2$ | Paralog$_1$) were removed from each dataset, so that only unique pairs (Paralog$_1$ | Paralog$_2$) were retained.

### Protein complexes resources
Protein Complexes definition were taken from Ori et al (2016). Members of protein complexes were mapped by orthology in *D. rerio* and *R. Norvegicus* using the bioconductor package "biomaRt" (Durinck et al, 2009) using as reference the *H. sapiens* protein complexes definitions.

### Publicly available data used in this study
Zebrafish embryo development data were obtained from White et al (2017) (Table S1). Human Proteome and Transcriptome data across tissues were obtained from Wang et al (2019) (Table S2). For these specific datasets, multi-mapping between reads was handled as it follows. For zebrafish embryo development data, the author used htseq-count to assign reads to its specific transcript. For this dataset reads that map to multiple genes were discarded. For human tissue atlas, Cufflinks v2.1.1 was used to assign reads to different transcript. In these cases multi-mapped reads were uniformingly split between genes. Protein identification and LFQ intensity values (Log$_2$) in cultured human iPSCs, NPCs and differentiated neurons, were obtained from Table S2 from Djuric et al (2017) and Frese et al (2017). Rat neuronal differentiation data published in Frese et al (2017), were downloaded from PRIDE (http://proteomecentral.proteomexchange.org/cgi/GetDataset?ID=PXD005031) and analyzed again as described below.

## Isolation of embryonic stem cells and neurons from zebrafish

Zebrafish (*D. rerio*) strains were maintained following standard protocols (Westerfield, 2007) in the Gilmour lab at the EMBL. Embryos were raised in E3 buffer (5 mM NaCl, 0.17 mM KCl, 0.33 mM CaCl$_2$, 0.33 mM MgSO$_4$) at 26–30°C. All zebrafish experiments were conducted on embryos younger than 3 d post fertilization. For isolation of undifferentiated cells a wild type strain (golden) and for neuronal cells the NBT-DsRed strain were used (Peri & Nüsslein-Volhard, 2008).

### Early embryos (6 hpf)
Wild type embryos were removed from their chorions using 1 ml of pronase (stock 30 mg/ml) in 40 ml buffer E3 and incubated for 10–15 min with gentle shaking every 2 min in a small beaker. The supernatant was removed and the embryos were washed four to five times using buffer E3. The embryos were splitted into batches of around 250–300 per 1.5 ml tube. 1 ml of deyolking buffer (55 mM NaCl, 1.8 mM KCl, 1.25 mM NaHCO$_3$) was added per tube and everything passed twice through a 200 $\mu$l pipet tip. The tubes were incubated at RT in a shaker at 1,100 rpm for 5 min and afterwards spun at 300$g$ for 30 s to remove the supernatant. The embryos were washed using 1 ml of wash buffer (110 mM NaCl, 3.5 mM KCl, 2.7 mM CaCl$_2$ and 10 mM Tris/HCl, pH 8.5), shaken at 1,100 rpm at RT for 2 min and spun as above to remove the supernatant. The wash step was repeated twice. The deyolked and dissociated embryos were resuspended in 400 $\mu$l wash buffer and passed through a 40 $\mu$m cell strainer to remove undissociated cells. The merged cells were washed as above and resuspended in 110 $\mu$l PBS and counted using a hemocytometer.

### Late embryos (24 hpf)
After 24 hpf, the NBT dsRed positive embryos were manually sorted. Up to the addition of the deyolking buffer all steps were the same as for early embryos. After the addition of 1 ml deyolking buffer per tube, the embryos were passed 10 times through a 1,000 $\mu$l pipet tip, followed by washing twice with deyolking buffer and four times with washing buffer. For better cell dissociation the embryos were rinsed once with Accumax (Millipore) and then resuspended in 1 ml

Accumax and transferred to a 15 ml tube. The embryos were incubated at RT for 5 min at the lowest speed of the vortex mixer. The embryos were dissociated by pipetting for 2 min using a 1,000 $\mu$l pipet tip, 2 min incubation on the vortex mixer and 1 min of additional pipetting. The cells were spun for 1 min at 300$g$ at RT and washed twice using 1 ml of PBS with 0.5% BSA. 400 $\mu$l of PBS with 0.5% BSA was used per tube to resuspend the cells afterwards passed through a 40 $\mu$m cell strainer and merged. DNAse I (10 mg/ml in water; Roche) 170 U/ml and 10 mM $MgCl_2$ was added. Cells expressing the DsRed fluorescent protein were FAC sorted with a MoFlo cell sorter (Beckman Coulter GmbH) to obtain a highly enriched fraction for neuronal cells.

### In vitro differentiation of mouse cortical neurons

#### Animal management practices

All mice were maintained in specific pathogen-free conditions, with food and water available ad libitum. The animal room had a constant temperature of 21°C ± 2°C, 55% ± 15% humidity, and controlled lighting (12 h light/dark cycle). The location for animal keeping was animal house TH4 at Leibniz Institute on Aging (Fritz Lipmann Institute). Breeding was license-free and performed under §11 TierSchG. Euthanasia and organ removal were performed under the internal §4 TierSchG licences O_CK_18-20 and O-CK_21-23. Euthanasia of mice was performed in a chamber with controlled CO2 fill rate according to "Directive 2010/63/EU annex IV of the European Parliament and the Council on the protection of animals used for specific purposes."

#### Mouse neuronal cell culture

Cortical neurons were isolated from wild type murine embryonic brains (E15.5) of mixed background (FVB/NJ, C57BL/6, 129/Sv) and differentiated in glia-conditioned neurobasal medium. Briefly, meninges were removed, cortices were isolated, minced and dissociated in trypsin EDTA (Invitrogen), solution for 15 min at 37°C. The supernatant was removed and the tissue was washed three times with trituration solution (10 mM Hepes, 1% penicillin/streptomycin, 10 mM L-glutamine, 1% BSA, 10% FBS, and 0.008% DNase in HBSS) and homogenized in trituration solution using fire polished glass pipettes. For the mouse in vitro neuronal differentiation data, neurons were counted and pellets containing 1 million cells (DIV0) were prepared and frozen until further use. In addition, 1 million cells were seeded on poly-L-lysine–coated 6-cm plates containing 4 ml glia-conditioned plating medium (1% penicillin/streptomycin, 1 mM sodium pyruvate, 0.5% glucose, 10 mM Hepes 1× B27 supplement, 10% FBS, and 10 mM L-glutamine in MEM). After 24 h, the plating medium was substituted by glia-conditioned neurobasal medium (10 mM Hepes, 1× B27 supplement, 5 mM L-glutamine in NBM). Neurons were collected at DIV3 and DIV10. To this end, neurons were scraped off in cold PBS and obtained cell suspensions were transferred to a microcentrifuge tube and centrifuged for 5 min at 4°C and 500$g$. The obtained pellets were washed with PBS twice and frozen until further use.

For preparation of glia-conditioned mediums, a primary astroglia culture was established. For this purpose, brains were isolated from 15.5 d old embryos, the meninges were removed, the cerebral hemispheres were minced and afterwards dissociated in trypsin

solution for 15 min at 37°C. Finally, the tissue was homogenized by pipetting and cells were plated on a 10 cm dish containing glia medium (1% penicillin/streptomycin, 1 mM sodium pyruvate, 0.5% glucose, 10 mM Hepes, 20 mM L-glutamine, and 10% FBS in MEM) and grown to confluence. For preconditioning of neurobasal medium or plating medium, the media were added to the glia feeder cultures and collected after 24 h.

#### Sec23a and Sec23b knockdown in mouse neuronal differentiation

For the Sec23 paralogs knockdowns, cortical neurons were isolated from C57BL/6JRj mouse embryo (Janvier), as described above. Then, freshly isolated neurons (5 million cells per nucleofection reaction) were transfected using the 4D-Nucleofector™ X Unit and the P3 Primary Cell 4D Nucleofector X kit (Lonza), as indicated. Cells were transfected with 250 nM of siRNA and 1 $\mu$l of control pMax GFP (Nucleofector X kit; Lonza), using the CU-133 program. Immediately after transfection, cells were plated on poly-L-lysine (Sigma-Aldrich)–coated 10-cm plates containing 10 ml of glia-conditioned plating medium: 1% penicillin/streptomycin, 1 mM sodium pyruvate (Sigma-Aldrich), 0.5% glucose, 10 mM Hepes, 1× B27 supplement (Invitrogen), 10% FBS, 10 mM L-glutamine in MEM (31095-052; Invitrogen), and incubated at 37°C. After 1 d, the medium was replaced with glia-conditioned neurobasal medium: 10 mM Hepes, 1× B27 supplement, 5 mM L-glutamine in NBM (Invitrogen). After 3 d in culture, neurons were washed twice with PBS, detached using Trypsin EDTA (3–5 min, 37°C), collected in 5 ml of PBS with 2% FBS, and pelleted by centrifugation (450$g$, 8 min, room temperature). Pellets were resuspended in 0.3 ml PBS with 2% FBS, and GFP-positive cells were labeled with Sytox Blue Dead Cell Stain (viable staining) (Molecular Probes; Thermo Fisher Scientific) and sorted directly in 200 $\mu$l of 2× lysis buffer (200 mM Hepes pH 8.0, 100 mM DTT, and 4% SDS) using a BD FACSAria Fusion with the Software BD FACSDiva 8.0.1 and 9.0.1 (BD Biosciences), using 488 nm laser and 530/30 filter for the GFP signal and laser 405 nm and 450/50 filter for the Sytox blue.

### Sample preparation for mass spectrometry

#### Sample preparation and dimethyl labeling for zebrafish stem cells and neurons

Cells were lysed by addition of Rapigest (Waters) and urea to a final concentration of 0.2% and 4 M, respectively, and sonicated for 3 × 30 s to shear chromatin. Before protein digestion, samples were stored at –80°C. Samples were quickly thawed and sonicated for 1 min. DTT was added to a final concentration of 10 mM and incubated for 30 min with mixing at 800 rpm to reduce cysteines. Then 15 mM of freshly prepared iodoacetamide (IAA) was added and samples were incubated for 30 min at room temperature in the dark to alkylate cysteines. Afterwards, 1:100 (w/w) LysC (Wako Chemicals GmbH) was added for 4 h at 37°C with mixing at 800 rpm. Then urea concentration was diluted to 1.5 M with HPLC water and 1:50 (w/w) trypsin (Promega GmbH) was added for 12 h at 37°C with mixing at 700 rpm. Afterwards, the samples were acidified with 10% TFA and the cleavage of Rapigest was allowed to proceed for 30 min at 37°C. After spinning the sample for 5 min at 130,00x $g$ at room temperature the supernatant was transferred to a new tube to proceed with peptide desalting. For desalting and cleaning-up of the

digested sample, C-18 spin columns (Sep-Pak C18 Classic Cartridge; Waters) were used. A vacuum manifold was used for all washing and elution steps. First, the columns were equilibrated with 100% methanol and then washed twice with 5% (vol/vol) acetonitrile (ACN) and 0.1% (vol/vol) formic acid (FA). The sample was loaded two times and then the column was washed two times with 5% (vol/vol) ACN and 0.1% (vol/vol) FA. The undifferentiated cell samples were labeled using a "light" labeling reagent and the FACS-sorted neuronal cells were labeled using an "intermediate" labeling reagent inducing a mass shift of 28 or 32 D, respectively (Boersema et al, 2009). Formaldehyde and the D-isotopomer of formaldehyde react with primary amines of peptides (N-terminus and side chains of lysines) and generate a mass shift of 4 D. The labeling reagents consisted of 4.5 ml 50 mM sodium phosphate buffer (mixture of 100 mM $NaH_2PO_4$ and 100 mM $Na_2HPO_4$), pH 7.5, 250 $\mu$l 600 mM NaBH3CN and 250 $\mu$l 4% formaldehyde for light or 4% deuterated formaldehyde for intermediate labeling reagent, per sample. After the labeling procedure, the column was washed three times with 5% (vol/vol) ACN and 0.1% (vol/vol) FA. For elution 50% (vol/vol) ACN and 0.1% (vol/vol) FA was used. Labeled peptides from undifferentiated cells and FACS sorted neurons were pooled, dried in a vacuum concentrator, and resuspended in 20 mM ammonium formate (pH 10.0), to be ready for high pH reverse-phase peptide fractionation. To dissolve the dried samples, they were vortexed, mixed for 5 min at maximum speed in a thermomixer and sonicated for 90 s. The samples were stored at –20°C.

### High pH reverse-phase peptide fractionation for dimethyl labeled samples

Offline high pH reverse-phase fractionation was performed using an Agilent 1200 Infinity HPLC System equipped with a quaternary pump, degasser, variable wavelength UV detector (set to 254 nm), peltier-cooled autosampler, and fraction collector (both set at 10°C). The column was a Gemini C18 column (3 $\mu$m, 110 Å, 100 × 1.0 mm; Phenomenex) with a Gemini C18, 4 × 2.0 mm SecurityGuard (Phenomenex) cartridge as a guard column. The solvent system consisted of 20 mM ammonium formate (pH 10.0) as mobile phase A and 100% acetonitrile as mobile phase B. The separation was accomplished at a mobile phase flow rate of 0.1 ml/min using the following linear gradient: 99% A for 2 min, from 99% A to 37.5% B in 61 min, to 85% B in a further 1 min, and held at 85% B for an additional 5 min, before returning to 99% A and re-equilibration for 18 min. Thirty two fractions were collected along with the LC separation that were subsequently pooled into 10 fractions. Pooled fractions were dried in a speed-vac and resuspended in 5% (vol/vol) ACN and 0.1% (vol/vol) FA and then stored at –80°C until LC–MS/MS analysis.

### Sample preparation for in vitro differentiated mouse neurons

Frozen cell pellets of in vitro differentiated mouse neurons (~1 million cells per sample) were thawed and resuspended in 100 $\mu$l of 1× PBS. An equivalent amount of 2× lysis buffer (200 mM Hepes pH 8.0, 100 mM DTT, 4% SDS) was added to the lysate, for a total volume of 200 $\mu$l. For neurons treated with *Sec23a*/b or control siRNA, cells (between 40,000 and 180,000 cells) were sorted directly into 2x lysis buffer. Samples were then sonicated in a Bioruptor Plus (Diagenode) for 10 cycles with 1 min ON and 30 s OFF with high intensity

at 20°C. Samples were then boiled for 10 min at 95°C, and a second sonication cycle was performed as described above. The lysates were centrifuged at 184,07$g$ for 1 min. Subsequently, samples were reduced using 10 mM DTT for 15 min at 45°C, and alkylated using freshly made 15 mM IAA for 30 min at room temperature in the dark. Subsequently, proteins were precipitated using acetone and digested using LysC (Wako sequencing grade) and trypsin (Promega sequencing grade), as described in Buczak et al (2020). The digested proteins were then acidified with 10% (vol/vol) trifluoroacetic acid. The eluates were dried down using a vacuum concentrator, and reconstituted samples in 5% (vol/vol) acetonitrile, 0.1% (vol/vol) formic acid. For data-independent acquisition (DIA)–based analysis (siRNA treated neurons), samples were transferred directly to an MS vial, diluted to a concentration of 1 $\mu$g/$\mu$l, and spiked with iRT kit peptides (Biognosys) before analysis by LC–MS/MS. For tandem mass tag (TMT)–based analysis (time course of in vitro differentiation), samples were further processed for TMT labeling as described below.

### TMT labeling and high pH reverse-phase peptide fractionation

After desalting, peptides were dried in a vacuum concentrator and buffered using 0.1M Hepes buffer, pH 8.5 (1:1 ratio), for labeling, and then sonicated in a Bioruptor Plus for five cycles with 1 min ON and 30 s OFF with high intensity. 10–20 $\mu$g peptides were taken for each labeling reaction. TMT-10plex reagents (Thermo Fisher Scientific) labeling was performed by addition of 1 $\mu$l of the TMT reagent. After 30 min of incubation at room temperature with shaking at 600 rpm in a thermomixer (Eppendorf), a second portion of TMT reagent (1 $\mu$l) was added and incubated for another 30 min. After checking labeling efficiency, samples were pooled, desalted with Oasis HLB $\mu$Elution Plate, and subjected to high pH fractionation before MS analysis. Offline high pH reverse-phase fractionation was performed using a Waters XBridge C18 column (3.5 $\mu$m, 100 × 1.0 mm; Waters) with a Gemini C18, 4 × 2.0 mm SecurityGuard (Phenomenex) cartridge as a guard column on an Agilent 1260 Infinity HPLC, as described in Buczak et al (2020). Forty-eight fractions were collected along with the LC separation, which were subsequently pooled into 16 fractions. Pooled fractions were dried in a vacuum concentrator and then stored at –80°C until LC–MS/MS analysis.

**Mass spectrometry data acquisition**

### Data-dependent acquisition for dimethyl labeled samples (zebrafish neurons and stem cells)

The 10 fractions obtained by high pH fractionation were analyzed using a nanoAcquity UPLC system (Waters GmbH) connected online to a LTQ-Orbitrap Velos Pro instrument (Thermo Fisher Scientific GmbH). Peptides were separated on a BEH300 C18 (75 $\mu$m × 250 mm, 1.7 $\mu$m) nanoAcquity UPLC column (Waters GmbH) using a stepwise 145 min gradient between 3% and 85% (vol/vol) ACN in 0.1% (vol/vol) FA. Data acquisition was performed using a TOP-20 strategy where survey MS scans (m/z range 375–1,600) were acquired in the Orbitrap (R = 30,000 FWHM) and up to 20 of the most abundant ions per full scan were fragmented by collision-induced dissociation (normalized collision energy = 35, activation Q = 0.250) and analyzed in the LTQ. Ion target values were 1 × 10$^6$ (or 500 ms maximum fill time) for full scans and 1 × 10$^5$ (or 50 ms maximum fill time) for

MS/MS scans. Charge states 1 and unknown were rejected. Dynamic exclusion was enabled with repeat count = 1, exclusion duration = 60 s, list size = 500 and mass window ±15 ppm.

### Data-dependent acquisition for TMT-labeled samples (mouse in vitro differentiation)

The 16 fractions obtained by high-pH fractionation were resuspended in 10 $\mu$l reconstitution buffer (5% [vol/vol] acetonitrile, 0.1% [vol/vol] TFA in water) and 3 $\mu$l were injected. Peptides were separated using the nanoAcquity UPLC system (Waters) fitted with a trapping (nanoAcquity Symmetry C18, 5 $\mu$m, 180 $\mu$m × 20 mm) and an analytical column (nanoAcquity BEH C18, 2.5 $\mu$m, and 75 $\mu$m × 250 mm). The outlet of the analytical column was coupled directly to an Orbitrap Fusion Lumos (Thermo Fisher Scientific) using the Proxeon nanospray source. Solvent A was water, 0.1% (vol/vol) formic acid, and solvent B was acetonitrile, 0.1% (vol/vol) formic acid. The samples were loaded with a constant flow of solvent A at 5 $\mu$l/min, onto the trapping column. Trapping time was 6 min. Peptides were eluted via the analytical column at a constant flow of 0.3 $\mu$l/min, at 40°C reconstitution buffer (5% [vol/vol] acetonitrile, 0.1% [vol/vol] TFA in water), and 3.5 $\mu$l were injected. Peptides were eluted using a linear gradient from 5 to 7% in 10 min, then from 7% B to 28% B in a further 105 min and to 45% B by 120 min. The peptides were introduced into the mass spectrometer via a Pico-Tip Emitter 360 $\mu$m OD ×20 $\mu$m ID; 10 $\mu$m tip (New Objective), and a spray voltage of 2.2 kV was applied. The capillary temperature was set at 300°C. Full-scan MS spectra with mass range 375–1,500 m/z were acquired in profile mode in the Orbitrap with resolution of 60,000 FWHM using the quad isolation. The RF on the ion funnel was set to 40%. The filling time was set at a maximum of 100 ms with an AGC target of 4 × 105 ions and 1 microscan. The peptide monoisotopic precursor selection was enabled along with relaxed restrictions if too few precursors were found. The most intense ions (instrument operated for a 3 s cycle time) from the full scan MS were selected for MS2, using quadrupole isolation and a window of 1 D. HCD was performed with collision energy of 35%. A maximum fill time of 50 ms for each precursor ion was set. MS2 data were acquired with a fixed first mass of 120 m/z. The dynamic exclusion list was with a maximum retention period of 60 s and relative mass window of 10 ppm. For the MS3, the precursor selection window was set to the range 400–2,000 m/z, with an exclude width of 18 m/z (high) and 5 m/z (low). The most intense fragments from the MS2 experiment were co-isolated (using Synchronus Precursor Selection = 8) and fragmented using HCD (65%). MS3 spectra were acquired in the Orbitrap over the mass range 100–1,000 m/z and resolution set to 30,000 FWMH. The maximum injection time was set to 105 ms, and the instrument was set not to injections for all available parallelizable time.

### Data-independent acquisition (Sec23a/b knockdowns)

Peptides were separated in trap/elute mode using the nanoAcquity MClass Ultra-High Performance Liquid Chromatography system (Waters; Waters Corporation) equipped with a trapping (nano-Acquity Symmetry C18, 5 $\mu$m, 180 $\mu$m × 20 mm) and an analytical column (nanoAcquity BEH C18, 1.7 $\mu$m, 75 $\mu$m × 250 mm). Solvent A was water and 0.1% formic acid, and solvent B was acetonitrile and 0.1% formic acid. 1 $\mu$l of the samples (~1 $\mu$g on column) were loaded

with a constant flow of solvent A at 5 $\mu$l/min onto the trapping column. Trapping time was 6 min. Peptides were eluted via the analytical column with a constant flow of 0.3 $\mu$l/min. During the elution, the percentage of solvent B increased in a nonlinear fashion from 0 to 40% in 120 min. Total run time was 145 min including equilibration and conditioning. The LC was coupled to an Orbitrap Exploris 480 (Thermo Fisher Scientific) using the Proxeon nanospray source. The peptides were introduced into the mass spectrometer via a Pico-Tip Emitter 360-$\mu$m outer diameter × 20-$\mu$m inner diameter, 10-$\mu$m tip (New Objective) heated at 300°C, and a spray voltage of 2.2 kV was applied. The capillary temperature was set at 300°C. The radio frequency ion funnel was set to 30%. For DIA data acquisition, full scan mass spectrometry (MS) spectra with mass range 350–1,650 m/z were acquired in profile mode in the Orbitrap with resolution of 120,000 FWHM. The default charge state was set to 3+. The filling time was set at a maximum of 60 ms with a limitation of 3 × $10^6$ ions. DIA scans were acquired with 40 mass window segments of differing widths across the MS1 mass range. Higher collisional dissociation fragmentation (stepped normalized collision energy; 25%, 27.5%, and 30%) was applied and MS/MS spectra were acquired with a resolution of 30,000 FWHM with a fixed first mass of 200 m/z after accumulation of 3 × $10^6$ ions or after filling time of 35 ms (whichever occurred first). Datas were acquired in profile mode. For data acquisition and processing of the raw data, Xcalibur 4.3 (Thermo Fisher Scientific) and Tune version 2.0 were used.

## Mass spectrometry data processing

### Data processing for dimethyl-labeled samples (zebrafish and rat neuronal differentiation)

Software MaxQuant (version 1.5.3.28) was used to search the MS.raw data. For *D. rerio*, the raw data were searched against the *D. rerio* UniProt database release: 2018_03, whereas for *R. norvegicus*, the .raw files from Frese et al (2017), were downloaded from PRIDE repository PXD005031 and searched against the UniProt *R. norvegicus* database release 2019_08. Both datasets were searched appending a list of common contaminants. The data were searched with the following modifications: Carbamidomethyl (C) (fixed) and Oxidation (M) and Acetyl (Protein N-term; variable). For *D. rerio* 2 labels, Light L (DmethLys0 and DmethNterm0) and Heavy H (DmethLys4 and DmethNterm4) were selected representing the stem cell and neurons, respectively. For the re-analysis of *R. norvegicus* data from Frese et al (2017), three different labels were used: Light L (DmethLys0 and DmethNterm0), Medium M, (DmethLys4 and DmethNterm4), and Heavy H (DmethLys8 and DmethNterm8). For identification, match between runs was selected with a match time window of 2 min, and an alignment time window of 20 min. The mass error tolerance for the full scan MS spectra was set at 20 ppm and for the MS/MS spectra at 0.5 D. A maximum of two missed cleavages was allowed. Identifications were filtered at 1% false discovery rate (FDR) at both peptide and protein levels using a target-decoy strategy (Elias & Gygi, 2007). From each experiment, iBAQ values (Schwanhäusser et al, 2011) and ratios between labels were extracted from the ProteinGroups.txt table. Differential expression analysis was performed using the mean of the normalized ratios between labels. The R package

fdrtool (Strimmer, 2008) was used to calculate *P*-values and q values for the different comparisons, on the Log$_2$ transformed mean ratios.

### Data processing for TMT10-plex data (mouse in vitro differentiation)

TMT-10plex data were processed using Proteome Discoverer v2.0 (Thermo Fisher Scientific). raw files were searched against the fasta database (UniProt *M. musculus* database, reviewed entry only, release 2016_11) using Mascot v2.5.1 (Matrix Science) with the following settings: Enzyme was set to trypsin, with up to one missed cleavage. MS1 mass tolerance was set to 10 ppm and MS2 to 0.5 D. Carbamidomethyl cysteine was set as a fixed modification, whereas oxidation of methionine and acetylation (N-term) were set as variable. Other modifications included the TMT-10plex modification from the quantification method used. The quantification method was set for reporter ions quantification with HCD and MS3 (mass tolerance, 20 ppm). FDR for peptide-spectrum matches (PSMs) was set to 0.01 using Percolator 13 (Brosch et al, 2009). Reporter ion intensity values for the PSMs were exported and processed with procedures written in R (v.4.0.5) and R studio server (v.1.2.5042 and 1.4.1106), as described in Heinze et al (2018). Briefly, PSMs mapping to reverse or contaminant hits, or having a Mascot score below 15, or having reporter ion intensities below 1 × 10$^3$ in all the relevant TMT channels were discarded. TMT channels intensities from the retained PSMs were then log$_2$ transformed, normalized and summarized into protein group quantities by taking the median value using MSnbase (Gatto & Lilley, 2012). At least two unique peptides per protein were required for the identification and only those peptides with no missing values across all 10 channels were considered for quantification. Protein differential expression was evaluated using the limma package (Ritchie et al, 2015). Differences in protein abundances were statistically determined using the *t* test moderated by the empirical Bayes method. *P*-values were adjusted for multiple testing using the Benjamini-Hochberg method (FDR, denoted as "adj. *P*") (Benjamini & Hochberg, 1995).

### Data processing for DIA samples (Sec23a/b knockdowns)

DIA libraries were created by searching the DIA runs using Spectronaut Pulsar (v13) and Biognosys. The data were searched against species specific protein databases (UniProt *M. musculus* release 2016_01) with a list of common contaminants appended. The data were searched with the following modifications: carbamidomethyl (C) as fixed modification, and oxidation (M), acetyl (protein N-term). A maximum of two missed cleavages was allowed. The library search was set to 1% FDR at both protein and peptide levels. Libraries contained a total of 101,659 precursors, corresponding to 5,708 and 6,003 protein groups, respectively. DIA data were then uploaded and searched against this spectral library using Spectronaut Professional (v.14.10) and default settings. Relative quantification was performed in Spectronaut for each pairwise comparison using the replicate samples from each condition using default settings, except: data filtering set to Qvalue sparse, and imputation to RunWise. Differential abundance testing was performed using a paired *t* test between replicates. The data (candidate tables) and protein quantity data reports were then exported for further data analyses.

### Data processing for human neuronal differentiation data

Protein identifications and LFQ intensity values (Log$_2$) in cultured iPSCs, NPCs and differentiated neurons, were obtained from the original Table S2 published in Djuric et al (2017). Differential expression analysis between the different conditions was performed on the log$_2$ LFQ intensity using the limma package (Ritchie et al, 2015).

## Data analysis

### Analysis of paralog pairs during development and across tissues

For the zebrafish development data (White et al, 2017), TPMs were used to calculate paralog pairs Pearson correlation coefficients. For the Human Tissue Atlas (Wang et al, 2019), Log$_2$(FPKM) and Log$_2$(IBAQ) were used to calculate correlation of paralog protein and transcript pairs. In all datasets, only genes and proteins identified in at least five time-points/tissues were considered for correlation analysis. Coefficient of variations ($\sigma$/mean protein or transcript expression along time points/tissues) was also calculated for every gene in each datasets. Genes that have at least one paralog in the genome according to Ensembl Compara were labeled as "Have Paralogs," and used for further analysis. From all the possible paralog pairs, three categories were created. The first one indicates all the possible paralog gene pairs, the second one indicates paralog pairs residing in the same protein complexes according to definitions from Ori et al (2016), and the third one given by the exclusion between the two, indicating all other paralog pairs, namely paralog pairs that do not reside in the same complexes. For every paralog pair, the mean sequence identity was then calculated as the mean reciprocal identity retrieved from the Ensembl database. The relationship between sequence identity and co-expression between paralog pairs, was evaluated using Pearson R correlation coefficient, and visualized through a Generalized Additive Model.

### Protein complex analysis during zebrafish development and across human tissues

For each datasets, proteins were annotated with the different protein complex definitions. Only protein complexes with at least five members present in each of the dataset were retained for analysis. For each of these complexes, all the possible pairwise correlations between complex members were considered, and from those the median value was used to calculate a median complex co-expression. We defined stable and variable complexes using the top and bottom 25% of the distribution, respectively. (1– median Pearson correlation) was also used to define then a measure of protein complex stoichiometric variability, as shown in Fig 1F and G. The distribution of correlations was then compared with a distribution of randomly assembled complexes of same size and complex members obtained by randomly assigning proteins/transcripts to complexes. For each dataset, the fraction of paralog pairs present was considered as the number of complex members that have at least one paralog in the genome divided by the total size of each protein complex. Finally for each subunit, we calculated expression correlation values with the other members of the same complex, taking the median of this value as a measure of co-expression for that specific subunit.

### Paralog regulation during neuronal differentiation

For each datasets, differentially expressed proteins between different conditions (Log$_2$ Fold-Change > 0.58 and adjusted *P*-value, or fdr tools *P*-value < 0.05) were selected. Proteins were annotated as "Have Paralogs" if they had at least one paralog annotated in the genome. For each comparison, we then considered all possible paralog pairs present in the data and identified unique paralog pairs that displayed concerted regulation (same Log$_2$ Fold Change sign for both paralogs) or opposite regulation (different Log$_2$ Fold Change sign).

### Complex members co-expression analysis for neuronal differentiation data

For calculating complex members stoichiometric variability, we adapted a previously established pipe-line (Gehring, 2021). For each condition and datasets, only protein complexes that had at least five quantified members were considered. Then for each subunit in each complex, the median euclidean distance of fold change between that subunit and all other complex members was calculated. The distance obtained was compared with a distribution of distances for 2,500 members from random complexes of equal size, obtained by randomly assigning proteins identified in the data to protein complexes. By comparing the two distributions we obtained a probability value for each subunit of observing lower distances with the complexes. Low *P*-values indicate high coexpression, denoted as stoichiometric stability, and vice versa.

### eggNOG mapping

Fasta proteomes sequences used for MS protein quantification of the different dataset were annotated using emapper-2.1.4-2 (Cantalapiedra et al, 2021), based on eggNOG orthology data (Huerta-Cepas et al, 2019). Sequence searches were performed using the software MMseqs2 (Steinegger & Söding, 2017). For each proteome, eggNOG annotation was performed using default parameters, using the Vertebrate level mapping.

### Conserved exchange of paralog proteins

For each dataset, protein quantification values were used to calculate paralog ratios across conditions. The log$_2$ paralog ratio between all possible quantified paralog pairs in each replicate was calculated for all the conditions tested. For each dataset, the significance of paralog ratio changes was assessed using the R package limma (Ritchie et al, 2015) considering replicates information. We considered only ratio changes relative to the first time point of each neuronal differentiation dataset. For comparison across species, each paralog pair was mapped to its relative eggNOG. Only paralog pairs where both entries could be mapped to a valid eggNOG were retained. After eggNOG mapping, shared eggNOG pairs between species were used to assess if specific paralog substitution were shared across different organisms, and for each specific comparison we combined the *P*-values using Fisher's combined probability test from the metaRNASeq R package (https://cran.r-project.org/web/packages/metaRNASeq/index.html). Combined *P*-values were corrected for multiple testing using the Benjamin-Hochberg correction (Benjamini & Hochberg, 1995). Because in some cases multiple proteins can map to the same eggNOG, for each pair and condition the mean value was considered for both ratio differences and *P*-values. From

this analysis, we considered as "conserved" only paralog gene pairs identified in all species and whose log$_2$ ratio changes were consistent in sign in at least five of the seven neuronal differentiation comparisons, with combined adjusted $P \leq 0.05$.

### Transcription factor enrichment analysis

Mouse promoter region sequences were obtained via the R package biomaRt (Durinck et al, 2009), selecting a 2,000-bp region upstream of the transcription starting site. TFs list were obtained from the JASPAR database (Fornes et al, 2020) via the TFBSTool BioconductoR package (Tan & Lenhard, 2016). Selecting a minimal score for enrichment of 80%. TFs that were not identified in a specific promoter were given an enrichment score of 0.

### GO enrichment analysis

Overrepresentation analysis of GO terms was performed with the R package topGO (Alexa & Rahnenfuhrer, 2021). Fisher test was used to estimate the expected proportion for different terms and obtain a *P*-value indicating the enrichment score for each specific GO term. Gene set enrichment analysis was performed with the topGO R package using a Kolmogorov–Smirnov test on the cumulative ranked distributions. For both enrichments *P*-values were adjusted using Hommel's correction, GO terms were considered significant if their adjusted *P*-values were below the value of 0.05. The R package rrvgo (https://ssayols.github.io/rrvgo/) was used to summarize and reduce redundancy of the enriched GO terms using default settings.

### Figure generation

Data visualization was performed with R (v.4.0.5) and R studio server (Version 1.4.1106) using the ggplot2 package (Wickham, 2009). Figs 1A, 2A, and 4A and graphical abstract were created with https://BioRender.com.

## Data Availability

### Mass spectrometry data repositories

Mass spectrometry proteomics data have been deposited to the ProteomeXchange Consortium via the PRIDE (Perez-Riverol et al, 2019) partner repository (https://www.ebi.ac.uk/pride/). Zebrafish neuronal differentiation with the dataset identifier PXD027191.

Mouse neuronal differentiation TMT10-plex are accessible with the identifier PXD027195. *Sec23a*, and *Sec23b* paralogs knockdowns in neurons DIA MS data are accessible with the identifier PXD027387.

### Code availability and analysis

A complete documentation of the code used for the analysis is available at https://genome.leibniz-fli.de/docs/paralogs/. A docker container with an interactive R studio session for replicating the analysis is available at https://cloud.leibniz-fli.de/index.php/s/MYEtyEJZ5LX7Yjp.

# Supplementary Information

# Acknowledgements

The authors gratefully acknowledge support from the Fritz Lipmann Institute (FLI) Core Facilities Proteomics, Flow Cytometry, Life Science Computing and Mouse, and the European Molecular Biology Laboratory Proteomics and FACS facilities. The authors thank Ivonne Heinze for processing samples for proteome analysis, Daniela Reichenbach, and Christina Valkova for assistance with neuronal mouse culture, and Andrea Gruia for support with fish care. A Ori acknowledges funding from the German Research Foundation (Deutsche Forschungsgemeinschaft, DFG) via the Research Training Group ProMoAge (GRK 2155), the Else Kröner Fresenius Stiftung (award number: 2019_A79), the Fritz-Thyssen foundation (award number: 10.20.1.022MN), and the Chan Zuckerberg Initiative Neurodegeneration Challenge Network (NDCN, award numbers: 2020-221617 and 2021-230967). The FLI is a member of the Leibniz Association and is financially supported by the Federal Government of Germany and the State of Thuringia. C. Kaether acknowledges funding from the German Research Foundation (Deutsche Forschungsgemeinschaft, DFG, grant number KA 1751/8-1). M. Helmer Citterich acknowledges funding from Fondazione AIRC per la Ricerca sul Cancro (AIRC, [IG 23539 to M Helmer Citterich]).

## Author Contributions

D Di Fraia: conceptualization, data curation, formal analysis, visualization, and writing—original draft.
M Anitei: methodology and writing—review and editing.
M-T Mackmull: methodology and writing—review and editing.
L Parca: conceptualization, data curation, supervision, and writing—review and editing.
L Behrendt: methodology and writing—review and editing.
A Andres-Pons: methodology.
D Gilmour: methodology.
M Helmer Citterich: conceptualization, supervision, funding acquisition, and writing—review and editing.
C Kaether: supervision, investigation, and writing—review and editing.
M Beck: conceptualization, supervision, project administration, and writing—review and editing.
A Ori: conceptualization, supervision, funding acquisition, investigation, writing—original draft, and project administration.

## Conflict of Interest Statement

The authors declare that they have no conflict of interest.

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
