## [Reviewer comments · Life Science Alliance]

Life Science Alliance

Conserved exchange of paralog proteins during neuronal differentiation

Domenico Di Fraia, Mihaela Anitei, Marie-Therese Mackmull, Luca Parca, Laura Behrendt, Amparo Andres-Pons, Darren Gilmour, Manuela Helmer-Citterich, Christoph Kaether, Martin Beck, and Alessandro Ori

DOI: <https://doi.org/10.26508/lsa.202201397>

Corresponding author(s): Alessandro Ori, Leibniz Institute on Aging - Fritz Lipmann Institute

Review Timeline:

Submission Date:	2022-02-01
Editorial Decision:	2022-02-01
Revision Received:	2022-02-14
Editorial Decision:	2022-02-14
Revision Received:	2022-02-16
Accepted:	2022-02-17

Transaction Report:

Please note that the manuscript was previously reviewed at another journal and the reports were taken into account in the decision-making process at *Life Science Alliance*.

Reviewers' Reports from another journal

Reviewer #1 Review

Report for Author:

The authors have carried out a comprehensive revision of the manuscript. They have added a number of additional analyses and figure panels and as a result I think the manuscript has been substantially improved. I particularly liked the new figure 2. All my other comments have been addressed as well, so I am happy to recommend publication of this manuscript.

Reviewer #2 Review

Report for Author:

Some of the aspects of the paper have been improved but some major issues are left and some more were added.

There are still some major concerns:

All comparisons are made on pairs of paralogs, some that are part of families. This means that pairwise values are not independent from each other while many of the statistical tests assume that the points are independent of each other, including the comparison of distributions.

The interpretation that paralogs are under pressure to diverge. Again, I assume that all pairwise comparisons are performed, which is incorrect. Ka/Ks values below 1 (0 on their plots) are not under pressure to diverge, they are under negative selection or purifying selection. There seems to be only a small fraction of values above 1 (0 on their plot), which means that there is very little selective pressure that leads to divergence. Some values are very high, probably around 10 or so. This looks like an artefact from the fact that old pairs are compared and Ks signal is saturated but not Ka . These estimates are unreliable. Finally, the choice of natural log to display this data is difficult to understand since most people cannot easily convert to linear scale easily. The biased estimation of Ka/Ks will depend on the age of the genes (Ks saturation) and we know from the other analyses

that the age of the genes correlate with some features. It is therefore not clear at all what these analyses tell us. The authors confirmed that some of their analyses (Cufflinks analysis) the reads that map to multiple targets are split equally between genes, which is incorrect. I understand that the authors show that proteomics data supports their analysis but I think this is not appropriate and it should have been redone without read splitting between pairs of paralogs. For instance, in a sample where one paralog is more expressed than the other, this even splitting would actually reduce their difference in expression, which is not something someone wants when performing analyses that try to understand how and why paralogs diverge.

Reviewer #1 Review

Report for Author:

In the manuscript "Conserved exchange of paralog proteins during neuronal differentiation" Di Fraia et al show that substitution of paralogous subunits may be an evolutionary conserved strategy for the functional diversification of protein complexes, which may be important for neurodevelopment. Overall, this is a well written manuscript about a well-planned and well-executed research project with an interesting research question. In general, the statistical analysis of the transcriptomics and proteomics data appears to be solid and the figures are clearly presented. I think this would be an important contribution to the field, but there are couple of major and some minor points that should be addressed first.

Major points

The title and abstract are written quite carefully and focus on the points that the manuscripts clearly delivers. Throughout the main text, however, the authors present (or imply) two additional concepts with potentially far-reaching consequences but these remain somewhat vague and appear to lack validation. I refer to whether or not (a) paralog switching is a genuine, common strategy for the developmental regulation of gene expression and (b) if protein complexes indeed play the main role in mediating this effect.

With regard to the first point, I wonder if they authors could show that paralogous genes are more likely to undergo developmentally regulated expression changes than other genes. In other words, can they rule out that this observation of "paralog switching" is the result of cherry-picking, i.e. focussing on a set of paralogs that just happen to be some of many expression changes accompanying neurodevelopment? Given that the authors have already assembled a powerful combination of transcriptomic and proteomic data sets they should be in a unique position to do this.

The second point relates to paralog co-regulation. I imagine that from a functional perspective there are different types of paralogs. Those that evolve to have complementary or redundant roles in the cell, which will be co-regulated. And those that substitute for one another to provide functional diversification within the same process, which will not be co-regulated. As the authors show, paralogs that are subunits of the same protein complex fall in the latter category. But is this specific to protein complexes, i.e. are divergently expressed paralogs actually enriched in complexes? Or are there many non-complex paralogs that are also divergently expressed, perhaps simply because they have different, competing functions?

Related to this, the enrichment of divergent paralogs in intracellular transport-related GO terms - could that just reflect a general enrichment of paralogs among transport-related proteins or a general enrichment of protein complexes with such functions? I guess what I'm asking is whether you can disentangle complex membership from divergent paralog behaviour, and if so, if the narrative of the paper should change accordingly.

Minor points

- Fig 1B: Show distribution for all genes (or non-paralogs) as a reference, to illustrate how much the paralog distribution deviates from this and to make sure that there are no normalisation issues with the data that could lead to an artificial enrichment of strong correlations
- Fig 1D/E: Very interesting observation, but if you talk about this panel in the main text using Pearson correlation coefficients it would be more consistent to show curves with a linear fit rather than a moving average
- Fig 1F/G: This relationship looks a little weak, possibly driven by a few hits that have a "fraction of 1" (are those heterodimers?) and might skew the picture a bit. It looks like there is no trend if those complexes are removed
- Fig S3: The GO plots are a bit messy. I think a table or bar chart would be easier to read and interpret
- Please explain what eggNOG is when you introduce it
- Fig 3: Does this still refer to complex subunits or paralogs in general?
- Fig 3B: I understand these changes are significant based on limma analysis, but they seem to be fairly small overall (~2-fold or less, or 4-fold if this is log₂ scale. Is it?). In that sense, is it correct to talk about a "switch" or "substitution" of paralogs? A 2-fold difference is more of a quantitative difference, i.e. some sort of functional fine-tuning, rather than a qualitative switch. Perhaps

this aspect could be discussed in the discussion.

- Fig 3B, part 2: Why are most changes positive? Did you select the order of the paralogs such that the ratio would be positive? Related to this, could you add a control panel showing paralogous pairs that are not developmentally regulated / do not significantly get "substituted" as a reference?
- Are there any hints as to which aspects of gene regulation may be responsible for differential paralog expression? Since the changes appear to be quite similar between the transcript and protein layers, one would suspect transcriptional differences. Alternatively, something that could be checked quite quickly, are the identified complexes enriched in non-exponentially degraded proteins (McShane et al, Cell 2016)? That would indicate that the regulation is due to differential complex incorporation and stabilisation of a paralog, and could explain how depletion of one paralog automatically upregulates the other one...

Reviewer #2 Review

Report for Author:

Di Fraia et al. examine the patterns of expression of paralogous genes in different species of vertebrates, including humans, mouse, rat, zebrafish. Some of the data they generated specifically for this study, other were publicly available. The authors find that paralogous genes have particularly variable patterns of expression across cell lines and across development, and do not always have positively correlated expression profiles. Indeed, some paralogs and negatively correlated expression profiles, which may suggest that they have functions that exclude each other.

From a conceptual point of view, this manuscript examines some questions that have been addressed in other papers and in different contexts, including some by this team. Indeed, the divergence of expression patterns among duplicated genes has been investigated in plants and animals in many papers. The enrichment of paralogous genes among those that display variable gene expression is always amply documented and this for many years.

Few examples:

<https://bmcpplantbiol.biomedcentral.com/articles/10.1186/s12870-020-02460-x>

<https://journals.plos.org/plosone/article?id=10.1371/journal.pone.0072362>

<https://genome.cshlp.org/content/14/10a/1870.full>

<https://pubmed.ncbi.nlm.nih.gov/28743766/>

<https://science.sciencemag.org/content/352/6288/1009.abstract>

<https://pubmed.ncbi.nlm.nih.gov/15122255/>

<https://www.pnas.org/content/106/7/2295>

The novelty is therefore limited because the authors do not go beyond the description of the diversity, with a few exceptions. For instance, there is no attempt to examine whether the age of the paralogs or the position of the paralogs in the genome is a significant factor. By comparing species for instance, it would be possible to have duplicates in human, mouse and rats that have a single copy in zebrafish because the duplication occurred after mammals originated, allowing to infer the ancestral state.

One of the interesting observations is that paralogs, which are known to be enriched in protein complexes (<https://genomebiology.biomedcentral.com/articles/10.1186/gb-2007-8-4-r51>, <https://pubmed.ncbi.nlm.nih.gov/17428571/>), are particularly variable in expression. The switching of paralogs has been reported in Ori et al. 2016. However, the analyses presented here do not go much further than that, apart from examining one particular pair of paralog of a protein complex functionally.

Overall, this is an interesting study but I believe it would require further work to make it a solid contribution to the field.

Specific comments:

Figure 1A: it would be useful to know how many samples and replicates are present in each dataset.

Figure 1E. It looks like the relationship is complex and it is not clear if the fit is done on a too small sample size so it is noisy or if this is real. Please show the datapoints.

Figures I, K, J, L: I am not sure how these examples were chosen. It looks like instead of having anticorrelated paralogs, there is always one highly expressed and one lowly expressed.

It is not clear if in the data analyses the RNAseq data was analysed in a way that takes into account cross mapping of reads between paralogs? If not, this could influence the correlation of expression across samples. If yes, this may lead to the elimination of many reads, which would increase the noise in the expression measurements and thus increase variation among samples, explaining some of the observations in Supp Fig1.

Are the paralog pairs considered exclusive paralogs (gene families of exactly 2) or larger families are considered. If yes, how do

the authors deal with the multiple pairwise comparisons within a family.

Line 282: what is a possible eggNOG?

Supp Fig3. The annotations in some of the plots cannot be read.

The experiments on knockdown are interesting but since the knockdown of one paralog leads to higher expression of the other, how do we know if the phenotypes measured are due to lower expression of one or higher expression of the other? Also, is SEC23B actually decreased in abundance upon SiSec23b treatment (Figure 4B)? Finally, if there is cross-regulation among paralogous pairs like we see partially here, it could explain why some pairs are negatively regulated in abundance. When one increases for some reasons, the other decreases. There would be no need for a more complex explanation.

Figure 4D: It is surprising that POU3F3 is significantly differentially expressed between SiCtr and SiSec23b. What is the statistical test used? Was there any correction for multiple testing?

A more general question is how do we know that the protein complex annotations that are used in this study are conserved across species? I know it would be impossible to have annotations that are species specific but this should be at least discussed and it should be shown that the assumption that the complexes have the same composition among species is supported by some data.

Some sections of the text could be improved. For instance, line 24, it is not clear what "general evolutionary pressure means". Some sentences seem to repeat earlier sections, for instance line 108, line 125, 138 repeat statements made earlier.

In general, the authors could use a more quantitative descriptions of the data. For instance, they say they look at whether some paralog substitutions are conserved. One would expect a certain degree of conservation which may depend on some features. There is not really a conserved versus non-conserved state, especially when we look at quantitative data.

Reviewer #3 Review

Report for Author:

Di Fraia et al., Conserved exchange of paralog proteins during neuronal differentiation

This paper studies the evolution of new functions by duplicate genes, by analyzing mRNA transcript and protein abundance data from several vertebrates (including, for various experiments, human, rat, mouse, and zebrafish, using a combination of new and previously published datasets) to assess the overall extent to which duplicate genes (paralogs) diverge in expression and participation in protein complexes. The authors then focus more specifically on neurogenesis and identify specific paralogs that appear to play different roles in differentiating neural tissue. The best characterized example presented is that of Sec23A/Sec23B, which the authors demonstrate appear to exchange for each other over the course of neural differentiation and in an evolutionarily conserved manner. Knockdown of the two Sec23 paralogs additionally produce opposing responses in in vitro neuronal differentiation, supporting functional divergence of the two otherwise highly similar proteins.

Overall, this is a nice study, with reasonably rigorous technical approaches (multiple quantitative proteomics strategies, appropriate care taken in gene orthology/paralogy calculations, and generally appropriate use of statistics, although see below about specific comments on the latter). I find the specific Sec23 result the most striking of the paper, especially the observation that knockdown of either Sec23A or B results in a compensatory increase in the abundance of the other subunit in order to maintain the same total abundance of Sec23A+B, yet the knockdowns exhibit markedly opposing effects on the cells' subsequent gene expression profiles. These data strongly support the authors' assertion that Sec23A/B do in fact have opposing roles with respect to neuronal differentiation, although the (direct) molecular mechanisms by which they carry out these roles are not yet known.

Major comments:

While I understand the statistical approach taken by the authors to prioritize a short list of paralogs that change roles during neurogenesis, I don't find the evidence for evolutionary conservation of the signal entirely clear-cut. It appears rather noisy for Sec23A/B and the other main examples chosen (Fig 3C, Supp Fig 5). I would surmise that much of this noise might arise from comparing different specific cell types, tissues, and developmental stages, which is of course a function of available datasets and intrinsic difficulties matching such measurements across species. Nonetheless, it would be nice for the authors to better address such issues in their discussion.

Along the same lines but more generally, I'm a bit concerned that the key analyses of paralogs during neurogenesis (Figs 2-3 and supporting) are underpowered by relying on only 2 to 3 abundance measurements, with replicates, per species. In such a case, I suspect that the authors require (not merely use) evolutionary conservation in order to help distinguish truly divergent cases from false positives. This is fine, as many other approaches exploit the same strategy, and the authors already point out

that "if some particular paralog substitutions are conserved across multiple organisms, they are more likely to functionally contribute to this process". However, it is important to acknowledge that the approach as implemented probably exhibits a high false negative detection rate. In fact, the terms "false negative" and "false positive" never appear in the paper with respect to identifying divergent paralogs. The authors should explicitly indicate for the neurogenesis section that the evolutionary signal is mostly used to improve the signal for detecting events (again, an entirely valid use), not to assess overall conservation of such events. Finally, they should more generally address (either by text or calculation) the issue of error rates accompanying their discovery of divergent paralogs.

Minor comments:

In the methods section, the authors should document which level of eggNOG mapping was used (e.g. eukaryotic, vertebrate, etc) which will obviously constrain which events it is possible to detect.

Authors' Response to Reviewers

We would like to thank the reviewers for their positive comments and constructive criticism that greatly helped us to improve our manuscript. Below we address point-by-point the comments of the reviewers.

Reviewer #1:

In the manuscript "Conserved exchange of paralog proteins during neuronal differentiation" Di Fraia et al show that substitution of paralogous subunits may be an evolutionary conserved strategy for the functional diversification of protein complexes, which may be important for neurodevelopment. Overall, this is a well written manuscript about a well-planned and well-executed research project with an interesting research question. In general, the statistical analysis of the transcriptomics and proteomics data appears to be solid and the figures are clearly presented. I think this would be an important contribution to the field, but there are couple of major and some minor points that should be addressed first.

We thank the reviewer for the positive comments!

Major points

The title and abstract are written quite carefully and focus on the points that the manuscript clearly delivers. Throughout the main text, however, the authors present (or imply) two additional concepts with potentially far-reaching consequences but these remain somewhat vague and appear to lack validation. I refer to whether or not (a) paralog switching is a genuine, common strategy for the developmental regulation of gene expression and (b) if protein complexes indeed play the main role in mediating this effect.

With regard to the first point, I wonder if they authors could show that paralogous genes are more likely to undergo developmentally regulated expression changes than other genes. In other words, can they rule out that this observation of "paralog switching" is the result of cherry-picking, i.e. focussing on a set of paralogs that just happen to be some of many expression changes accompanying neurodevelopment? Given that the authors have already assembled a powerful combination of transcriptomic and proteomic data sets they should be in a unique position to do this.

We thank the reviewer for pointing this out. Indeed, we observe that genes that have paralogs are more likely to undergo variations in their expression profiles, both during development and across tissues (see Appendix Figure S1 panel B-D). Similar results are also shown during the neurogenesis process across multiple species, where paralog genes tend to have higher fold-change values compared to genes that do not have paralog in the genome (see Figure 4 panel B). Thus we believe that expression variability is indeed a characteristic of paralog genes.

(Line 119-123, Page 3) - “By calculating coefficient of variations for each protein and transcript, we also noticed that genes that possess paralogs in the genome tend to be more variably expressed during development (Appendix Fig S1B) (Wilcoxon test $p < 2.2E-16$), and across differentiated tissues both at the transcriptome Appendix Fig S1C) (Wilcoxon test $p < 2.2E-16$) and proteome level (Appendix Fig S1D) (Wilcoxon test $p < 2.2E-16$).”

Appendix Figure S1 - Paralogs contribute to transcriptome and proteome variability during development and across tissues

B-D - Coefficient of variation of genes that have paralogs (blue) and genes that do not have any (grey). Density distributions are shown for Zebrafish development (B), transcript across tissues (C) and protein across tissues (D).

(Line 279-281, Page 7)“The neuronal differentiation data recapitulated the general patterns of paralog expression that we observed during development and across tissues: (i) proteins that have at least one paralog in the genome displayed larger fold changes (Fig 4B)█

Figure 4 - Changes of abundance of paralog proteins during neuronal differentiation

B - Boxplots display absolute \log_2 fold changes during neuronal differentiation for proteins that have (blue) or do not have (grey) at least one paralog.

The second point relates to paralog co-regulation. I imagine that from a functional perspective there are different types of paralogs. Those that evolve to have complementary or redundant roles in the cell, which will be co-regulated. And those that substitute for one another to provide functional diversification within the same process, which will not be co-regulated. As the authors show, paralogs that are subunits of the same protein complex fall in the latter category. But is this specific to protein complexes, i.e. are divergently expressed paralogs actually enriched in complexes? Or are there many non-complex paralogs that are also divergently expressed, perhaps simply because they have different, competing functions?

We thank the reviewer for raising this point. We analyzed the enrichment of divergent paralogs in protein complexes and found modest, but significant differences in the proportion of divergent paralog pairs (lower 25% of the total correlation distribution) between paralog pairs that are part of the same protein complex, and other paralog pairs. Specifically, we found a slightly higher proportion of divergent paralogs in non complex members at least across human tissues (Fig. R1). No significant differences were retrieved in the zebrafish development data. This indicates that alternative usage of paralogs extend beyond protein complexes. We have clarified this in the revised manuscript:

(Line 125-127, Page 3) - “Even though alternative paralog usage extends beyond protein complexes, it is already known that substitution of paralog members can contribute to the functional specialization of large protein complexes”

Figure R1 - Paralogs divergence is not restricted to protein complexes
 Stacked bar plot showing the percentage of paralog pairs that are divergent (bottom 25% of the correlation distribution, grey) and all other paralog pairs (blue). Paralog pairs are divided on the x-axis according to the fact that they are members or not of the same protein complex. Asterisks indicate p values of the Fisher test between the two compared groups: ** $p \leq 0.01$, **** $p \leq 0.0001$, ns = not significant.

Additionally, we have included a new analysis section where we explored the relationship between paralog co-expression and the evolutionary pressure for paralog pairs to diverge in sequence. We indeed found that paralogs that tend to have an evolutionary pressure to diverge in sequence (indicative of functional diversification) are also generally more concerted in their expression levels, and vice versa (Fig 3D).

Figure 3 - Evolutionary pressure for paralog pairs to diverge in their co-expression

D - Boxplot showing the relation between paralog pairs regulation profile and their evolutionary pressure to diverge in sequence (expressed as $\text{Logn}(Ka/Ks)$) for both paralog pairs that are part of the same complex and paralog pairs that do not. (Anti-regulated = lower 25% of the distribution, Other = >25% and <75% of the distribution, Co-regulated = top 25% of the distribution). Data for transcriptomes across human tissues. Asterisks indicate p values of the two-sided Wilcoxon test between the two compared groups: * $p \leq 0.05$; ** $p \leq 0.01$, *** $p \leq 0.001$, **** $p \leq 0.0001$, ns = not significant

(Line 224-229, Page 5): *“We speculate that, in general, a higher evolutionary pressure to diverge in sequence allows paralogs to be co-expressed by enabling them to achieve different functions. On the other hand, a subset of paralogs that are constrained to maintain a certain degree of sequence identity (lower Ka / Ks ratios) tend to be expressed in a mutually exclusive fashion, presumably in order to be able to carry out similar functions, but to fine tune them in a context/cell-type specific manner”* .

Related to this, the enrichment of divergent paralogs in intracellular transport-related GO terms - *could that just reflect a general enrichment of paralogs among transport-related proteins or a general enrichment of protein complexes with such functions?* I guess what I'm asking is whether you can disentangle complex membership from divergent paralog behaviour, and if so, if the narrative of the paper should change accordingly.

We thank the reviewer for the comments. Indeed our Gene Ontology Enrichments are carried out just between genes that have paralogs, and thus any paralog specific functions are normalized in reference to the Gene Ontology analysis. In the specific case of Figure 1H, the GOTerm overrepresentation analysis is carried out only at the level of paralog pairs that are part of protein complexes. In this sense, the specific enrichment of transport-related proteins is then a characteristic of paralogs that have variable expression behaviour. We apologize for the lack of clarity regarding this point in the original manuscript. In the revised

manuscript, we have clarified this aspect by better explaining how GO enrichments were carried out.

(Line 153-156, Page 4) - “To gain insight into which biological functions are carried out by paralog pairs that show divergent expression profiles, we performed a GO Term overrepresentation analysis. We found that, within protein complexes, the anti-correlated paralog pairs (bottom 25% of the distribution) are enriched in terms related to vesicle mediated transport (Fig 1H, Dataset EV3).”

Minor points

- Fig 1B: Show distribution for all genes (or non-paralogs) as a reference, to illustrate how much the paralog distribution deviates from this and to make sure that there are no normalisation issues with the data that could lead to an artificial enrichment of strong correlations

We thank the reviewer for this comment. In order to address this point, we have included for comparison a distribution of correlations obtained from random gene pairs. Indeed, we show that paralog pairs show distributions of correlations that are significantly different from random gene pairs. We have included this comparison in Fig. 1B and C.

Figure 1 - Expression of paralog genes during Zebrafish development and across human tissues

B/C - Density distribution of paralog gene pairs. Pearson correlations during zebrafish embryo development (B) and across human tissues (C). Colored areas highlight different correlation intervals. Labels indicate the percentage of paralogs that are positively correlated ($R > 0$) and negatively correlated ($R \leq 0$). Barplot indicates the number of paralog pairs present in each category. Transcriptome data were used for both comparisons. Dashed red lines indicate correlation distribution of random gene pairs.

• Fig 1D/E: Very interesting observation, but if you talk about this panel in the main text using Pearson correlation coefficients it would be more consistent to show curves with a linear fit rather than a moving average

We thank the reviewer for pointing this out. In order to avoid any bias due to the fitting function used or small sample size, we decided instead to bin paralogs on the basis of their reciprocal sequence identity and analyse for each bin the distribution of the observed co-expression correlations for both complex members and other complexes. For transparency, we also indicated the number of paralog pairs in each bin for both groups (Fig. 1D and E). This new analysis reproduced our previous observation for both the development and tissue dataset, namely that paralog pairs residing in the same complex, tend to have (in respect of their sequence identity) lower co-expression values compared to other paralog pairs.

Figure 1. Expression of paralog genes during Zebrafish development and across human tissues

D/E - Boxplots representing transcriptome paralog pairs Pearson correlation distributions, in relationship to different reciprocal identity intervals for zebrafish development (D) and human tissues (E). Colors indicate paralog pairs that are members of the same protein complex (blue), and all other paralog pairs (red). Asterisks indicate p values of the two-sided Wilcoxon test between the two compared groups: * $p \leq 0.05$; ** $p \leq 0.01$, *** $p \leq 0.001$, **** $p \leq 0.0001$.

• Fig 1F/G: This relationship looks a little weak, possibly driven by a few hits that have a "fraction of 1" (are those heterodimers?) and might skew the picture a bit. It looks like there is no trend if those complexes are removed

We thank the reviewer for pointing this out. In order to address this point, we have repeated the analysis by removing heterodimers and complexes composed only of paralog genes. After removal of these complexes, the positive correlation between paralog content and complex variability remains significant. We have included this additional analysis as Figure EV1C and mention this additional control in the main text.

(Line 144-147, Page 4) - "A similar pattern can be observed using transcriptome data from human tissues ($R=0.23$, $p = 7.9E-05$) (Fig 1G, Dataset EV2). Consistent results were also

obtained when heterodimers and complexes composed only of paralog genes were excluded from the analysis (Fig EV1C).

Figure EV1. Paralogs contribute to the variability of protein complexes

C - Relationship between paralogs content (fraction of subunits that have paralogs in the genome) and complex stoichiometric variability, in cases where heterodimers and complexes composed only of paralogs were removed from the analysis. Complex stoichiometric variability is expressed as 1-R, where R is the median Pearson correlation of expression between all complex subunits.

• Fig S3: The GO plots are a bit messy. I think a table or bar chart would be easier to read and interpret

We have updated the visualization of the GO plot as suggested in the revised Figure EV2.

Figure EV 2. Proteome data of neuronal differentiation

A-D - [...] a barplot displaying the over representation analysis for 'Biological Process' GO terms enriched among upregulated proteins (Log2 fold change ≥ 0.58) against the rest of the quantified proteins is shown. In each plot, the y axis indicates the $-\log_{10}(\text{adjusted } p \text{ value})$ of the Fisher test. DIV = differentiation *in vitro* day, iPSC=induced pluripotent stem cell, Neu = Neurons, NPC = neuronal precursor cell, Stem = undifferentiated stem cell.

- Please explain what eggNOG is when you introduce it

We have included a short explanation of eggNOG in the result paragraph, as suggested:

(Line 296-299, Page 7) - *“To compare paralog pairs across species, we took advantage of the eggNOG resource (Huerta-Cepas et al. 2019). Using the eggNOG pipeline, we annotated each paralog to its orthology group (eggNOG) enabling consistent comparison of paralog genes across species.”*

- Fig 3: Does this still refer to complex subunits or paralogs in general?

Figure 3 refers to paralog in general. We have now made this more clear in the text:

(Line 299-301, Page 7) - *“We calculated abundance ratios for all the possible paralog eggNOG pairs across conditions and we assessed significant changes in these ratios using a linear model..“*

- Fig 3B: I understand these changes are significant based on limma analysis, but they seem to be fairly small overall (~2-fold or less, or 4-fold if this is log2 scale. Is it?). In that sense, is it correct to talk about a "switch" or "substitution" of paralogs? A 2-fold difference is more of a quantitative difference, i.e. some sort of functional fine-tuning, rather than a qualitative switch. Perhaps this aspect could be discussed in the discussion.

We thank the reviewer for raising this important point. Indeed, we observe a relatively broad range of effect sizes and in most cases both paralogs remain expressed, but their relative proportion changes as a consequence of opposite regulation. We fully agree that this rather reflects a potential fine-tuning, e.g., of protein complex functions. We have now included in the discussion paragraph a passage that clarifies our definition of paralog substitution to reflect this important considerations.

(Line 433-438, Page 10) - *“The divergent expression of paralog pairs that we describe implies changes of their relative abundances across developmental stages / cell types. We define such changes as paralog “substitutions” or “exchanges”. However, we observed a broad range of effect sizes and, in multiple cases, both paralogs remain expressed. We believe that such substitutions reflect rather a fine-tuning than a qualitative switch of, e.g., protein complex function.”*

- Fig 3B, part 2: Why are most changes positive? Did you select the order of the paralogs such that the ratio would be positive? Related to this, could you add a control panel showing paralogous pairs that are not developmentally regulated / do not significantly get "substituted" as a reference?

In order to compare ratio differences between paralog across dataset, we have arbitrarily decided to prioritize positive ratios. For this reason, most of the changes in the heatmap appear as positive since they are conserved across datasets. A few exceptions show

opposite ratio differences and therefore appear negative. We have clarified this arbitrary choice in the figure legend.

(Line 649-650, Page 14) - “In order to compare ratio differences between paralogs across dataset, positive ratios were arbitrarily prioritized.”

In addition, we have included a control panel as suggested by the reviewer in Appendix Figure S4.

Appendix Figure S4. Paralogs pairs ratio changes across dataset

A - Histogram showing the $-\log_{10}(p)$ values distribution for changes in ratio between paralog pairs across all the proteomic dataset considered in the study. Dashed line represents the $p = 0.05$ significance cut-off. Labels display the percentage of the significant changes in ratio between paralog pairs in the different datasets.

B - Upset plot showing the number of shared significant changes in paralog ratio between datasets. Only changes in ratio shared in at least 2 species are shown in the plot.

- Are there any hints as to which aspects of gene regulation may be responsible for differential paralog expression? Since the changes appear to be quite similar between the transcript and protein layers, one would suspect transcriptional differences. Alternatively,

something that could be checked quite quickly, are the identified complexes enriched in non-exponentially degraded proteins (McShane et al, Cell 2016)? That would indicate that the regulation is due to differential complex incorporation and stabilisation of a paralog, and could explain how depletion of one paralog automatically upregulates the other one...

We thank the reviewer for raising this important point. In the revised manuscript, we have more directly compared differential paralog expression at the transcriptome and proteome level for the human tissues dataset. Indeed, we observed a significant positive correlation between transcript and protein level changes indicating a substantial contribution of transcriptional regulation (Fig 2B). However, we also identified a subset of paralog that diverge in expression changes between transcriptome and proteome (Fig 2D). Therefore, we followed the suggestion of the reviewer and analysed paralog co-expression in relation to their degradation profiles according to (McShane et al, Cell 2016). We found that pairs that include at least one non-exponentially degraded protein (NED) typically display stronger co-regulation at the transcriptome than at their proteome level (Fig 2D). This suggests that indeed non-exponential protein degradation might contribute to determining the relative levels of this subset of paralogs, independently of transcriptional regulation.

Figure 2. Transcriptional regulation and protein degradation determine the relative levels of paralog pairs

A - Violin plot showing distribution of Pearson correlation values between transcriptome and proteome across human tissues for genes that have paralogs (blue) and genes that do not (grey) . Asterisks indicate p values of the Wilcoxon test between the compared groups: **** $p < 0.0001$.

B - Hexbin scatterplot showing the relationship between paralog pairs co-expression (expressed as R Pearson correlation between paralog pairs) at both transcriptome (x-axis) and proteome (y-axis). Color scale indicates paralog pair count in each of the represented bins of the plot.

C - Barplot showing the proportion of not-exponentially degraded proteins (NED, green) and exponentially degraded proteins (ED, orange) for proteins that have paralogs and that are either part of protein complexes or not. Asterisks indicate p values of the Fisher test between the compared groups: *** $p < 0.001$, ns=not significant.

D - (Left panel) Scatterplot showing the relationship between paralog pairs co-expression (expressed as R Pearson correlation between paralog pairs) at both transcriptome (x-axis) and proteome (y-axis). Color scale indicates “delta” score (differences in paralog pair correlation between transcriptome and proteome). Paralog pairs more co-expressed at transcript are represented in blue, while paralog pairs more coexpressed at the proteome are represented in red. (Right panel) Boxplot showing the distribution of differences between proteome and transcriptome paralog pairs co-expression in relationship to their degradation profile as calculated in McShane et al., 2016. Color ruler bar indicates differences in correlation between proteome and transcript as expressed in the left panel. Asterisks indicate p values of the two-sided Wilcoxon test between the two compared groups: **** $p < 0.0001$, ns = not significant.

Reviewer

#2:

Di Fraia et al. examine the patterns of expression of paralogous genes in different species of vertebrates, including humans, mouse, rat, zebrafish. Some of the data they generated specifically for this study, others were publicly available. The authors find that paralogous genes have particularly variable patterns of expression across cell lines and across development, and do not always have positively correlated expression profiles. Indeed, some paralogs and negatively correlated expression profiles, which may suggest that they have functions that exclude each other.

From a conceptual point of view, this manuscript examines some questions that have been addressed in other papers and in different contexts, including some by this team. Indeed, the divergence of expression patterns among duplicated genes has been investigated in plants and animals in many papers. The enrichment of paralogous genes among those that display variable gene expression is always amply documented and this for many years.

Few examples:

<https://bmcpplantbiol.biomedcentral.com/articles/10.1186/s12870-020-02460-x>

<https://journals.plos.org/plosone/article?id=10.1371/journal.pone.0072362>

<https://genome.cshlp.org/content/14/10a/1870.full>

<https://pubmed.ncbi.nlm.nih.gov/28743766/>

<https://science.sciencemag.org/content/352/6288/1009.abstract>

<https://pubmed.ncbi.nlm.nih.gov/15122255/>

<https://www.pnas.org/content/106/7/2295>

The novelty is therefore limited because the authors do not go beyond the description of the diversity, with a few exceptions. For instance, there is no attempt to examine whether the age of the paralogs or the position of the paralogs in the genome is a significant factor. By comparing species for instance, it would be possible to have duplicates in human, mouse and rats that have a single copy in zebrafish because the duplication occurred after mammals originated, allowing us to infer the ancestral state.

We thank the reviewer for the constructive criticism and the suggestions on how to expand the scope of our manuscript. In the revised manuscript, we included further analysis aimed at (i) exploring the mechanisms regulating the relative levels of paralog pairs (Fig. 2, see reply to Reviewer 1); (ii) investigating the relationship between paralog regulation, time of divergence and evolutionary pressure to maintain sequence identity; (iii) assessing the transcriptional regulation of paralog expression in the context of neurogenesis, specifically the presence of specific transcription factor binding motifs in up-stream regions of paralogs that show divergent expression.

First, we investigated the relationship between the co-expression of paralog pairs and the relative evolutionary constraints to which they are subjected to:

(Line 203-244, Page 5) *“For each paralog pair, we calculated Ka / Ks ratio on the respective cDNA sequences.*

We focused on the transcriptome atlas of human tissues and we observed a negative relationship between co-expression of paralog pairs and both Ks and Ka, ($R=-0.29$, $p < 2.20E-16$ and $R=-0.26$, $p < 2.20E-16$ respectively, Fig3A, Fig3B), confirming that paralog gene expression tends to diverge with time (Guschanski, Warnefors, and Kaessmann 2017). This metric also consistently indicates that paralog pairs that are part of the same protein complexes, tend to display, on average, lower Ka / Ks ratios compared to other paralogs (Wilcoxon test pvalue $< 2.20E-16$) Fig3C, likely due to the existence of an evolutionary pressure to maintain a certain degree of sequence identity between complex members, owing to structural constraints necessary for complex assembly.

Interestingly, when we analyzed the relationship between Ka / Ks ratio and paralog pairs co-expression, we found that co-expressed paralogs tend to display a higher evolutionary pressure to diverge in sequence (higher Ka / Ks ratios), as compared to paralogs that show opposite expression profiles. This relationship is true for both protein complex members and other paralogs (Fig3D, Wilcoxon test p value = $4.00E-03$ and $3.80E-08$, respectively). We speculate that, in general, a higher evolutionary pressure to diverge in sequence allows paralogs to be co-expressed by enabling them to achieve different functions. On the other hand, a subset of paralogs that are constrained to maintain a certain degree of sequence identity (lower Ka / Ks ratios) tend to be expressed in a mutually exclusive fashion, presumably in order to be able to carry out similar functions, but to fine tune them in a context/cell-type specific manner .

Lastly we took into consideration time of divergence, expressed as the last common ancestor between two paralog pairs. We found again a negative correlation between paralog transcriptome co-expression, time of divergence and sequence identity. Specifically, more

ancient paralogs tend to display lower levels of co-expression as compared to more recent ones (Fig 3E). Surprisingly, this relationship had a different trend for members of protein complexes (Fig 3E). We observed a sharp increase in sequence identity together with a decrease of both correlation and pressure to diverge (Ka/Ks ratio) for those paralogs that are members of protein complexes and duplicated after the origin of vertebrates (Fig 3E). Keeping in mind limitations due to the relatively small number of paralog pairs analysed for certain groups, these metrics might indicate an expansion of protein complex modularity that originated with the vertebrate lineage. Intriguingly, GO terms overrepresentation analysis on these specific genes against all other paralogs pairs in protein complexes showed an enrichment for functions related to vesicle trafficking, similarly to what we observed for paralogs that diverge in expression during development and in human tissues (Appendix Fig S2A)”

Figure 3. Evolutionary pressure for paralog pairs to diverge in their co-expression

A - Scatter plot showing the relation between a linear transformation for paralog pairs correlation, expressed as $\log_n((1+R)(1-R))$ (where R = pearson correlation coefficient between paralog pairs) , and the rate of synonymous mutation (Ks) between paralog pairs. Orange dashed line indicates the regression line for the relation. Data for transcriptomes across human tissues.

B - Scatter plot showing the relation between a linear transformation for paralog pairs correlation, expressed as $\log_n((1+R)(1-R))$ (where R = pearson correlation coefficient of paralog pairs) , and the rate of non-synonymous mutation (Ka) between paralog pairs. Orange dashed line indicates the regression line for the relation. Data for transcriptomes across human tissues.

C - Violin plot showing the distribution of $\log_n(Ka/Ks)$ ratio for paralog pairs that take part in the formation of the same protein complex (blue) and all other paralog pairs (red). Asterisks indicate p values of the Wilcoxon test between the compared groups: **** $p \leq 0.0001$.

D - Boxplot showing the relation between paralog pairs regulation profile and their evolutionary pressure to diverge in sequence (expressed as $\log_n(Ka/Ks)$) for both paralog pairs that are part of the same complex and paralog pairs that do not. (Anti-regulated = lower 25% of the distribution, Other = >25% and <75% of the distribution, Co-regulated = top 25% of the distribution). Data for

transcriptomes across human tissues. Asterisks indicate p values of the two-sided Wilcoxon test between the two compared groups: * p<= 0.05; ** p<=0.01, *** p<=0.001, **** p<=0.0001, ns = not significant

E - Relationship between paralog pairs last common ancestor with *Homo sapiens* (y-axis) and Pearson R correlation (top), mean reciprocal sequence identity (middle) and Logn(Ka/Ks), for paralog pairs that are members of the same protein complex (blue) and all other paralog pairs (red). Dark-colored lines indicate the median value of each bin, while shaded area indicates the middle 50% of the distribution. Number on the y-axis indicates the number of paralog pairs present in each category.

Appendix Figure S2. GOEnrichment Analysis of paralog pairs that are complex members in relationship with their last common ancestor with human

A - Barplots illustrating the results of a GOTerm overrepresentation analysis (ORA) for paralog pairs that are also protein complex members, divided by their last common ancestor with *Homo sapiens*. x-axis indicates the $-\log_{10}$ (adjusted p value) of the Fisher enrichment test. GOTerms were summarized in order to reduce redundancy (see methods section for details).

Second, we investigated differences in the genomic context of paralog genes that might contribute to their regulation during neurogenesis:

We investigated transcription factor binding motifs in the regulatory elements of paralog pairs showing divergent expression during neuronal differentiation.

(Line 329-346, Page 8) “Using the mouse genome as a reference, we searched for transcription factor binding motifs in a 2000bp region upstream of the transcription starting site (TSS). For each of the conserved paralog pairs we then quantified the percentage of shared transcription factor binding sites. We found that overall conserved paralog pairs shared a high percentage of regulatory elements (68%), however a subset of transcription factor binding motifs (32%) could be identified only in one of the two paralogs.

We then calculated for each regulatory element differences in transcription factor (TF) enrichment score between paralog pairs Fig6A. Among regulatory elements that showed greater difference in enrichment score between paralog pairs (TF identified in at least 10 of the conserved paralog pairs), we found binding motifs for multiple retinoic-acid receptors (Rara, Rarb, Rxrb, Rxra, Rxrg) and transcription factors known to be involved in neuronal differentiation (Sox21, Sox1) Fig 6B

We observed that groups of paralog pairs shared enrichment for specific regulatory regions Fig6C., indicating the possibility of a common regulation of paralog pairs by the activity of specific transcription factors, especially for the ones related to transport and redox processes. Together these data suggest that differential transcriptional regulation of paralog genes by specific transcription factors can explain, at least in part, the paralog substitutions observed during neuronal differentiation.”

Figure 6. Differential Transcription factors binding site between paralog pairs promoters

A - Barplot showing differences between transcription factors score binding site, for different paralog gene pairs (colored in blue and red). Top 5 most different transcription factors in score are shown as labels.

B - Distribution of differences in score between paralog pairs for different transcription factors. Top 20 (in terms of their median difference) transcription factors identified in at least 10 of the conserved paralog pairs are shown. Red dots indicate the median of the distribution. Grey segment indicates the 25% till the 75% of the distribution. Transcription factors related to neuronal differentiation and retinoic acid signalling are highlighted in bold.

C - Network visualization for example transcription factors related to neuronal differentiation and retinoic acid signalling, linked to the paralog in which their binding site was identified in a 2000bp promoter region. Different paralog genes related to transport (orange) and redox (blue) are affected.

One of the interesting observations is that paralogs, which are known to be enriched in protein complexes (<https://genomebiology.biomedcentral.com/articles/10.1186/gb-2007-8-4-r51>, <https://pubmed.ncbi.nlm.nih.gov/17428571/>), are particularly variable in expression. The switching of paralogs has been reported in Ori et al. 2016. However, the analyses presented

here do not go much further than that, apart from examining one particular pair of paralog of a protein complex functionally.

We politely disagree with the reviewer regarding this point. Our manuscript expands on previous work by (i) investigating the conservation across species of the divergent expression of paralog genes in the context of a well defined cell differentiation process, namely neurogenesis, (ii) identifying a subset of paralogs involved in the transport of macromolecules that are particularly prone to undergo substitution during neuronal differentiation, and (iii) experimentally validating the relevance of one of such substitutions, namely the exchange between the paralogs SEC23A and B. As mentioned above, in the revised manuscript we have expanded further our investigation by assessing the mechanisms regulating the relative levels of paralogs, studying the relationship between sequence conservation and paralog co-expression, and identifying transcription factors that differentially regulate specific paralogs and thereby might contribute to the paralog exchanges that we describe to occur during neurogenesis.

Overall, this is an interesting study but I believe it would require further work to make it a solid contribution to the field.

Specific comments:

Figure 1A: it would be useful to know how many samples and replicates are present in each dataset.

We included this information in Figure 1A and Figure 4A of the revised manuscript.

Figure 1. Expression of paralog genes during Zebrafish development and across human tissues

A - Transcriptome data during zebrafish embryo development (White et al. 2017) and transcriptome and proteome data from 29 healthy human tissues (Wang et al. 2019) were used to calculate Pearson correlation of expression during development and across tissues for paralog gene pairs. Pie plots indicate the proportion of quantified transcripts that possess at least one paralog in the zebrafish and human dataset, respectively.

Figure 4. Changes of abundance of paralog proteins during neuronal differentiation

A - Overview of dataset used and data analysis workflow. DIV = differentiation *in vitro* day, iPSC=induced pluripotent stem cell, Neu = Neurons, NPC = neuronal precursor cell, Stem = undifferentiated stem cell.

Figure 1E. It looks like the relationship is complex and it is not clear if the fit is done on a too small sample size so it is noisy or if this is real. Please show the datapoints.

We thank the reviewer for pointing this out. In order to avoid any bias due to the fitting function used or small sample size, we decided instead to bin paralog pairs on the basis of their reciprocal sequence identity and analyse for each bin the distribution of the observed co-expression correlations for both complex members and other complexes. For transparency, we also indicated the number of paralog pairs in each bin for both groups (Fig. 1D and E). This new analysis reproduced our previous observation for both the development and tissue dataset, namely that paralog pairs residing in the same complex, tend to have (in respect of their sequence identity) lower co-expression values compared to other paralog pairs.

Figure 1. Expression of paralog genes during Zebrafish development and across human tissues

D/E - Boxplots representing transcriptome paralog pairs Pearson correlation distributions, in relationship to different reciprocal identity intervals for zebrafish development (D) and human tissues (E). Colors indicate paralog pairs that are members of the same protein complex (blue), and all other paralog pairs (red). Asterisks indicate p values of the two-sided Wilcoxon test between the two compared groups: * $p < 0.05$; ** $p < 0.01$; *** $p < 0.001$; **** $p < 0.0001$.

Figures I, K, J, L: I am not sure how these examples were chosen. It looks like instead of having anticorrelated paralogs, there is always one highly expressed and one lowly expressed.

We thank the reviewer for pointing this out and we apologize for the lack of clarity. All the plots display on the y-axis log₂ fold changes relatively to day 0 of development and show opposite expression patterns of paralog pairs with one paralog displaying increased expression (positive fold change) and the other one decreased expression (negative fold change). The divergent expression patterns lead to a negative correlation of the expression profiles between the two paralogs as shown in Figure 1B. These examples were chosen among members of protein complexes that showed the most extreme divergent expression profiles. To better relate Figures 1I-L to the rest of the Figure, we have now indicated in each panel the correlation values between the expression profiles.

Figure 1 - Expression of paralog genes during Zebrafish development and across human tissues

I-L - Transcriptome profiles along embryo development for specific paralog pairs part of chromatin organization complexes, BAF/SWI (K) and HBO1 (L), or vesicle-transport complexes, COPII (I) and COPI (J). Log₂ fold changes calculated from TPMs relatively to the first time point are shown.

It is not clear if in the data analyses the RNAseq data was analysed in a way that takes into account cross mapping of reads between paralogs? If not, this could influence the correlation of expression across samples. If yes, this may lead to the elimination of many reads, which would increase the noise in the expression measurements and thus increase variation among samples, explaining some of the observations in Supp Fig1.

We thank the reviewer for raising this point. The RNA-Seq data from zebrafish development were analyzed by the authors using the software htseq-count from the htseq (Anders, Pyl, and Huber 2015) suite program. The algorithm deals with reads that map to multiple genes by simply discarding them. While this approach can lead in some cases to an

underestimation of the quantification for specific genes, it also ensures more precision and reduces uncertainty in the quantification of very similar genes (such as paralogs).

The RNA-Seq data from the human tissue atlas were instead analyzed by the authors using Cufflinks v2.1.1. Cufflinks deal with multiple reads by uniformly splitting them equally between genes (Trapnell et al. 2010; Deschamps-Francoeur, Simoneau, and Scott 2020). This strategy ensures that every read gets counted only once, thus reducing redundancy, and also ensuring reduction in variability. However, as pointed out by the reviewer, the splitting of common reads could influence the correlation of expression for paralog gene pairs. On the other hand, the quantification of the corresponding proteome data from the same tissues was based on proteotypic (unique) peptides, which are by definition paralog-specific. Despite these technical differences, the correlation coefficients between expression profiles of paralog pairs obtained from transcript or protein levels showed comparable distributions (compare Fig.1C and Appendix Figure S1), and they were positively and significantly correlated between each other (Fig. 2B).

Therefore, based on the consistency of our findings across datasets and, within the human dataset, across transcriptome and proteome layers, we conclude that patterns of co-expression of paralog genes that we describe are not due to technical artifacts but rather reflect a biological characteristic of these specific genes.

For transparency, we have included these technical informations regarding the processing of the RNAseq dataset in the Methods section.

(Line 814-819, Page 18) - *“For these specific datasets, multi-mapping between reads was handled as follows. For Zebrafish Embryo development data, the author used htseq-count to assign reads to its specific transcript. For this dataset reads that map to multiple genes were discarded. For human tissue atlas, Cufflinks v2.1.1 was used to assign reads to different transcripts. In this case multi-mapped reads were reads by uniformly splitted between genes.”*

Are the paralog pairs considered exclusive paralogs (gene families of exactly 2) or larger families are considered. If yes, how do the authors deal with the multiple pairwise comparisons within a family.

We also considered larger families, in this case we did not apply any specific constraints. We just consider all the possible paralog pairs in the dataset. We have stated this aspect more clearly in the revised manuscript.

(Line 108-109, Page 3) - *“We used correlation analysis of transcripts and proteins encoded by all possible paralog gene pairs to address their co-regulation during development and in fully differentiated tissues”*

Line 282: what is a possible eggNOG?

We have included a short explanation of eggNOG in the result paragraph, as suggested.

(Line 296-299, Page 7) - "To compare paralog pairs across species, we took advantage of the eggNOG resource(Huerta-Cepas et al. 2019). Using the eggNOG pipeline, we annotated each paralog to its orthology group (eggNOG) enabling consistent comparison of paralog genes across species"

Supp Fig3. The annotations in some of the plots cannot be read.

We have improved the visualization of the annotations as suggested in Figure EV2.

Figure EV2. Proteome data of neuronal differentiation

A-D - [...] a barplot displaying the over representation analysis for 'Biological Process' GO terms enriched among upregulated proteins (\log_2 fold change ≥ 0.58) against the rest of the quantified proteins is shown. In each plot, the y axis indicates the $-\log_{10}(\text{adjusted p value})$ of the Fisher test. DIV = differentiation *in vitro* day, iPSC=induced pluripotent stem cell, Neu = Neurons, NPC = neuronal precursor cell, Stem = undifferentiated stem cell.

The experiments on knockdown are interesting but since the knockdown of one paralog leads to higher expression of the other, how do we know if the phenotypes measured are due to lower expression of one or higher expression of the other?

We thank the reviewer for this comment. Indeed our data show that knock-down of either of the two SEC23 paralogs influence the level of the other paralog. We believe this is part of a post-transcriptional mechanism that ensures the maintenance of the correct levels of the SEC23 paralogs relative to other COPII components. Accordingly, we show that the summed abundance of the SEC23, although significantly changing, shows a rather small effect size ($<15\%$ increase in the siSec23b condition) (Appendix Fig S5A), thus excluding that the phenotypes that we observed are due to depletion of the SEC23 pool. Importantly, what is affected by the different siRNA treatment is the ratio between SEC23A and SEC23B (Fig 7B). We thus conclude that the different proteome signatures that we observed correlate

with a different proportion between SEC23A and SEC23B, similarly to what is observed during neuronal differentiation.

Appendix Figure S5 - Effects of Sec23a/b knockdowns

A - Barplot shows the summed Log₂ protein quantities of Sec23a and Sec23b following different siRNA treatment. Asterisks indicate significance for the paired two-sample Wilcoxon tests between conditions. ** p<=0.01, ns = not significant. n=6.

Figure 7 - Altering the ratio between Sec23a and Sec23b affects neuronal differentiation *in vitro*.

B - Protein abundance of Sec23a (green) and Sec23b (orange) following different siRNA treatments, estimated from mass spectrometry data. n=6.

Also, is SEC23B actually decreased in abundance upon SiSec23b treatment (Figure 4B)?

Yes, SEC23B is significantly decreased upon siSec23b (log₂FC siSec23B vs. siCtrl= -0.42 , Qvalue = 1.05 E-04). Importantly, as mentioned above, the compensatory increase in SEC23A leads to an inversion in the abundance ratio of the two paralogs between the condition siSec23A and siSec23B (Fig. 7B).

Finally, if there is cross-regulation among paralogous pairs like we see partially here, it could explain why some pairs are negatively regulated in abundance. When one increases for some reason, the other decreases. There would be no need for a more complex explanation.

We thank the reviewer for this comment. Indeed, we believe this is one of the possible mechanisms that regulate relative levels of paralog pairs. We have addressed this potential mechanism of paralog regulation more systematically in Figure 2 (see also reply to Reviewer 1).

(Line 170-197, Page 4/5) *“In order to understand which mechanisms contribute to concerted or divergent paralog regulation, we first analyzed the correlation between transcript and protein expression profiles across human tissues for individual genes. We found that genes that have at least one paralog in the genome display, on average, higher concordance between transcript and protein levels compared to genes that do not. This was true for both paralogs that are members of protein complexes (Wilcoxon test pvalue < 2.20E-16 ,Fig 2A) and other paralogs (Wilcoxon test pvalue < 2.20E-16 ,Fig 2A), and it indicates a substantial transcriptional control of paralog protein levels in human tissues. Consistently, co-expression profiles of paralog pairs are generally positively and significantly correlated at the transcriptome and proteome level (R=0.39, $p < 2.20E-16$,Fig 2B).*

To assess the contribution of additional post-transcriptional mechanisms, we took advantage of a proteome kinetic analysis of protein degradation profiles(McShane et al. 2016). This work defined two major protein degradation patterns in human cells, namely proteins that are exponentially degraded (ED) and proteins that exhibit an initial rapid degradation upon synthesis followed by relatively stable levels (non-exponentially degraded, NED). At first, we noted that paralogs that are members of protein complexes are enriched in NEDs compared to other paralogs (Fisher test pvalue = 4.34E-14) Fig 2C, in agreement with what was already observed for protein complex members in general (McShane et al. 2016).

Next, we assessed the relationship between protein degradation profiles and the concordance of transcript and protein co-expression for paralog pairs. To do so, we calculated a “delta” score between the pairwise correlations of paralogs at the protein and transcript levels. Positive delta scores indicate paralog pairs that show higher co-expression at the protein level and, conversely, negative delta scores point to co-expression at the transcript level but divergent expression at the protein level. We found that pairs that include at least one NED paralog tend to display significantly lower “delta” scores as compared to pairs where both proteins are ED (Wilcoxon test pvalue = 3.20E-08 , and pvalue= 2.41E-11 Fig 2D), suggesting that non-exponential protein degradation might contribute to determine the relative levels of this subset of paralogs, independently of transcriptional regulation.”

Figure 2. Transcriptional regulation and protein degradation determine the relative levels of paralog pairs

A - Violin plot showing distribution of Pearson correlation values between transcriptome and proteome across human tissues for genes that have paralogs (blue) and genes that do not (grey) . Asterisks indicate p values of the Wilcoxon test between the compared groups: **** $p \leq 0.0001$.

B - Hexbin scatterplot showing the relationship between paralog pairs co-expression (expressed as R Pearson correlation between paralog pairs) at both transcriptome (x-axis) and proteome (y-axis). Color scale indicates paralog pair count in each of the represented bins of the plot.

C - Barplot showing the proportion of not-exponentially degraded proteins (NED, green) and exponentially degraded proteins (ED, orange) for proteins that have paralogs and that are either part of protein complexes or not. Asterisks indicate p values of the Fisher test between the compared groups: *** $p \leq 0.001$, ns=not significant.

D - (Left panel) Scatterplot showing the relationship between paralog pairs co-expression (expressed as R Pearson correlation between paralog pairs) at both transcriptome (x-axis) and proteome (y-axis). Color scale indicates “delta” score (differences in paralog pair correlation between transcriptome and proteome). Paralog pairs more co-expressed at transcript are represented in blue, while paralog pairs more coexpressed at the proteome are represented in red. (Right panel) Boxplot showing the distribution of differences between proteome and transcriptome paralog pairs co-expression in relationship to their degradation profile as calculated in McShane et al., 2016. Color ruler bar indicates differences in correlation between proteome and transcript as expressed in the left panel. Asterisks indicate p values of the two-sided Wilcoxon test between the two compared groups: **** $p \leq 0.0001$, ns = not significant.

Figure 4D: It is surprising that POU3F3 is significantly differentially expressed between SiCtr and SiSec23b. What is the statistical test used? Was there any correction for multiple testing?

We thank the reviewer for pointing this out. Differential expression of proteins for this experiment was assessed using Spectronaut. Spectronaut performs pairwise t-test for each protein using intensity values of all the precursors identified for a given protein group. The t-tests are then corrected for multiple testing for the whole experiment (Q values). For display purposes, we plotted the summarized protein intensities derived from the median value of all the precursors for each condition. We have realized that in the original manuscript some of the examples chosen had been selected using a filter based on unadjusted p value. We apologize for this mistake and we have now updated the manuscript by selecting candidates that are significant after multiple testing corrections. We have more clearly explained how the statistics and plotted values were obtained in the relevant figure legend of the revised manuscript.

(Line 683-684, Page 15) - *“Asterisk indicates p values from a paired t-test run at the precursor level and corrected from multiple testing as implemented in the Spectronaut software”*

(Line 373-382, Page 9) - *“Among these proteins, Sec23b knockdown decreased the amount of Bub1b, an essential component of the mitotic checkpoint (Chan et al. 1999), as well as Notch2 a well-known regulator of cell-fate determination and known to inhibit differentiation of cerebellar neuron precursors (Solecki et al. 2001). On the other hand it increased the levels of Synaptotagmin-1 (Syt1), a neuronal synaptic protein involved in neurotransmitter release (Coppola et al. 2001) (Fig7D). Instead, knock-down of Sec23a increased the expression of the transcription factor Pou3f3 that has been shown to be necessary for the earliest state of neurogenesis (Sugitani 2002; Dominguez, Ayoub, and Rakic 2013) and, relatively to Sec23b-KD, of the component of the COP9 signalosome (Myod, also known as Cops9) that has been described to promote proliferation (Denti et al. 2006) (Fig 7E)”*

A more general question is how do we know that the protein complex annotations that are used in this study are conserved across species? I know it would be impossible to have annotations that are species specific but this should be at least discussed and it should be shown that the assumption that the complexes have the same composition among species is supported by some data.

We thank the reviewer for raising this important point. In order to assess the validity of our complex definitions for different species, we have analysed the co-expression of members of the same complex across datasets. We and others have previously shown that members of protein complexes are significantly more co-expressed than other protein pairs (Liu, Yuan, and Li 2009; Ori et al. 2016; Romanov et al. 2019). Consistently and across all the dataset analysed, we found protein complex members (according to the definition used) to display significantly higher correlation than random pairs of proteins. We thus conclude that to a large extent our complex definitions can be applied to different species. We have included this important control analysis in the revised manuscript (Appendix FigS3).

Appendix Figure S3 - Protein Complex co-expression across datasets

A - Density distribution of co-expression of protein complexes across the different proteomic datasets considered in the study. Complex co-expression is indicated as (1-p), where p = probability of observing lower fold-change distances between protein complex subunits (see methods for detail). The distribution density highlights in green the different protein complex definition as used in our study, against a random set of protein complexes (grey, red dashed) obtained by randomly assembling protein complexes of equal size. Red p values indicate the p values resulted from a two-sided Wilcoxon test performed on the two distributions.

Some sections of the text could be improved. For instance, line 24, it is not clear what "general evolutionary pressure means". Some sentences seem to repeat earlier sections, for instance line 108, line 125, 138 repeat statements made earlier.

We thank the reviewer for pointing this out. We have updated accordingly sections in our manuscript to minimize redundancy, in particular:

(Line 23-24, Page 1) - *“Gene duplication enables the emergence of new functions by lowering the evolutionary pressure that is posed on the ancestral genes”.*

(Line 138-139, Page 3, unrevisited manuscript) - *“In order to estimate the contribution of these paralog genes to context-dependent protein complex formation, we..”*

(Line 162-263, Page 4, unrevisited manuscript) - *“suggesting a potential divergent role of paralog proteins in establishing or modulating these biological functions”.*

In general, the authors could use a more quantitative descriptions of the data. For instance, they say they look at whether some paralog substitutions are conserved. One would expect a certain degree of conservation which may depend on some features. There is not really a conserved versus non-conserved state, especially when we look at quantitative data.

We thank the reviewer for this comment. We fully agree that there are some limitations to our definition of conserved paralog exchanges. Some of these limitations might arise from the heterogeneity of experimental design and consequent statistical power of the different studies analysed (as pointed out by Reviewer 1), but also from the way we decided to define an exchange as conserved or not. We decided to consider a substitution as “conserved” by applying a stringent cut-off: Log2 paralog ratio differences consistent in direction in all species and at least in 5 of the 7 conditions considered, with a combined p value < 0.05.

In order to be more transparent on the overlap between dataset, we have included an upset plot in the revised manuscript that shows the intersections between all the dataset used (Appendix Fig S4B). In addition, we have included in the revised manuscript a paragraph that describes more clearly the potential limitations of our study in the Discussion section.

(Line 445-452, Page 10) - *“By integrating proteomic datasets from different species, we have identified patterns of paralog regulation that occur during neuronal differentiation in multiple vertebrates. Despite heterogeneity in the cell types and developmental stages compared as well as technical differences between dataset that might have limited our ability to accurately quantify specific paralogs, [1] we were able to extract a signature of paralog substitutions based on the detection of consistent abundance changes across species. The enrichment of paralogs involved in the transport of macromolecules supports the hypothesis of fine tuning of membrane trafficking-related functions during neuronal differentiation”*

Appendix Figure S4. Paralogs pairs ratio changes across dataset

A - Histogram showing the $-\log_{10}(p \text{ values})$ distribution for changes in ratio between paralog pairs across all the proteomic dataset considered in the study. Dashed line represents the $p = 0.05$ significance cut-off. Labels display the percentage of the significant changes in ratio between paralog pairs in the different datasets.

B - Upset plot showing the number of shared significant changes in paralog ratio between datasets. Only changes in ratio shared in at least 2 species are shown in the plot.

Di Fraia et al., Conserved exchange of paralog proteins during neuronal differentiation

This paper studies the evolution of new functions by duplicate genes, by analyzing mRNA transcript and protein abundance data from several vertebrates (including, for various experiments, human, rat, mouse, and zebrafish, using a combination of new and previously published datasets) to assess the overall extent to which duplicate genes (paralogs) diverge in expression and participation in protein complexes. The authors then focus more specifically on neurogenesis and identify specific paralogs that appear to play different roles in differentiating neural tissue. The best characterized example presented is that of Sec23A/Sec23B, which the authors demonstrate appear to exchange for each other over the course of neural differentiation and in an evolutionarily conserved manner. Knockdown of the two Sec23 paralogs additionally produce opposing responses in in vitro neuronal differentiation, supporting functional divergence of the two otherwise highly similar proteins.

Overall, this is a nice study, with reasonably rigorous technical approaches (multiple quantitative proteomics strategies, appropriate care taken in gene orthology/paralogy calculations, and generally appropriate use of statistics, although see below about specific comments on the latter). I find the specific Sec23 result the most striking of the paper, especially the observation that knockdown of either Sec23A or B results in a compensatory increase in the abundance of the other subunit in order to maintain the same total abundance of Sec23A+B, yet the knockdowns exhibit markedly opposing effects on the cells' subsequent gene expression profiles. These data strongly support the authors' assertion that Sec23A/B do in fact have opposing roles with respect to neuronal differentiation, although the (direct) molecular mechanisms by which they carry out these roles are not yet known.

We thank the reviewer for the positive comments on our manuscript!

Major comments:

While I understand the statistical approach taken by the authors to prioritize a short list of paralogs that change roles during neurogenesis, I don't find the evidence for evolutionary conservation of the signal entirely clear-cut. It appears rather noisy for Sec23A/B and the other main examples chosen (Fig 3C, Supp Fig 5). I would surmise that much of this noise might arise from comparing different specific cell types, tissues, and developmental stages, which is of course a function of available datasets and intrinsic difficulties matching such measurements across species. Nonetheless, it would be nice for the authors to better address such issues in their discussion.

We thank the reviewer for raising this important point. Indeed, heterogeneity between dataset both biological and technical, e.g., due to experimental design (see also below), might limit our ability to detect conserved signatures of paralog exchanges. As suggested, we have now addressed this important potential limitation of our study in the discussion paragraph.

(Line 433-438, Page 10) "The divergent expression of paralog pairs that we describe implies changes of their relative abundances across developmental stages / cell types. We define

such changes as paralog “substitutions” or “exchanges”. However, we observed a broad range of effect sizes and, in multiple cases, both paralogs remain expressed. We believe that such substitutions reflect rather a fine-tuning than a qualitative switch of, e.g., protein complex function.”

Along the same lines but more generally, I'm a bit concerned that the key analyses of paralogs during neurogenesis (Figs 2-3 and supporting) are underpowered by relying on only 2 to 3 abundance measurements, with replicates, per species. In such a case, I suspect that the authors require (not merely use) evolutionary conservation in order to help distinguish truly divergent cases from false positives. This is fine, as many other approaches exploit the same strategy, and the authors already point out that "if some particular paralog substitutions are conserved across multiple organisms, they are more likely to functionally contribute to this process". However, it is important to acknowledge that the approach as implemented probably exhibits a high false negative detection rate. In fact, the terms "false negative" and "false positive" never appear in the paper with respect to identifying divergent paralogs. The authors should explicitly indicate for the neurogenesis section that the evolutionary signal is mostly used to improve the signal for detecting events (again, an entirely valid use), not to assess overall conservation of such events. Finally, they should more generally address (either by text or calculation) the issue of error rates accompanying their discovery of divergent paralogs.

This is also an extremely valid point. We have now included an analysis of the pairwise overlap of detected paralog exchanges between dataset. It can be noted that in some dataset, e.g. mouse TMT dataset, many more paralog substitutions can be detected than in other dataset (Appendix Figure S4B). We believe that this reflects technical differences in the analysed dataset (e.g., depth of analysis in terms of the number of precursors per protein identified) that influence the ability to accurately quantify the abundance of specific paralogs. We have included this analysis in the revised manuscript and modified the text as suggested by the Reviewer.

Appendix Figure S4. Paralogs pairs ratio changes across dataset

A - Histogram showing the $-\log_{10}(p \text{ values})$ distribution for changes in ratio between paralog pairs across all the proteomic dataset considered in the study. Dashed line represents the $p = 0.05$ significance cut-off. Labels display the percentage of the significant changes in ratio between paralog pairs in the different datasets.

B - Upset plot showing the number of shared significant changes in paralog ratio between datasets. Only changes in ratio shared in at least 2 species are shown in the plot.

(Line 305-311, Page 7) - *“At first we noticed differences in the number of paralog pairs displaying significant changes of ratios in the compared datasets, with the mouse dataset showing the largest number of detected changes (Appendix Fig S4A). We speculate that these differences might be related to heterogeneity in the proteomic workflows and experimental designs used across studies leading to different proteome coverages and limited statistical power of some of the datasets. Despite this limitation, we could identify subsets of paralog ratio changes that were common to at least two species (Appendix Fig S4B)”*

Minor comments:

In the methods section, the authors should document which level of eggNOG mapping was used (e.g. eukaryotic, vertebrate, etc) which will obviously constrain which events it is possible to detect.

We have clarified this important detail in the method section of the revised manuscript.

(Line 1235-1236, Page 27) - *“For each proteome eggNOG annotation was performed using default parameters, using the Vertebrate level mapping”*

Bibliography

- Anders, Simon, Paul Theodor Pyl, and Wolfgang Huber. 2015. "HTSeq--a Python Framework to Work with High-Throughput Sequencing Data." *Bioinformatics* 31 (2): 166–69.
- Deschamps-Francoeur, Gabrielle, Joël Simoneau, and Michelle S. Scott. 2020. "Handling Multi-Mapped Reads in RNA-Seq." *Computational and Structural Biotechnology Journal* 18 (June): 1569–76.
- Guschanski, Katerina, Maria Warnefors, and Henrik Kaessmann. 2017. "The Evolution of Duplicate Gene Expression in Mammalian Organs." *Genome Research* 27 (9): 1461–74.
- Huerta-Cepas, Jaime, Damian Szklarczyk, Davide Heller, Ana Hernández-Plaza, Sofia K. Forslund, Helen Cook, Daniel R. Mende, et al. 2019. "eggNOG 5.0: A Hierarchical, Functionally and Phylogenetically Annotated Orthology Resource Based on 5090 Organisms and 2502 Viruses." *Nucleic Acids Research*. <https://doi.org/10.1093/nar/gky1085>.
- Liu, Ching-Ti, Shinsheng Yuan, and Ker-Chau Li. 2009. "Patterns of Co-Expression for Protein Complexes by Size in *Saccharomyces Cerevisiae*." *Nucleic Acids Research* 37 (2): 526–32.
- McShane, Erik, Celine Sin, Henrik Zauber, Jonathan N. Wells, Neysan Donnelly, Xi Wang, Jingyi Hou, et al. 2016. "Kinetic Analysis of Protein Stability Reveals Age-Dependent Degradation." *Cell* 167 (3): 803–15.e21.
- Ori, Alessandro, Murat Iskar, Katarzyna Buczak, Panagiotis Kastiris, Luca Parca, Amparo Andrés-Pons, Stephan Singer, Peer Bork, and Martin Beck. 2016. "Spatiotemporal Variation of Mammalian Protein Complex Stoichiometries." *Genome Biology* 17 (March): 47.
- Romanov, Natalie, Michael Kuhn, Ruedi Aebersold, Alessandro Ori, Martin Beck, and Peer Bork. 2019. "Disentangling Genetic and Environmental Effects on the Proteotypes of Individuals." *Cell* 177 (5): 1308–18.e10.
- Trapnell, Cole, Brian A. Williams, Geo Pertea, Ali Mortazavi, Gordon Kwan, Marijke J. van Baren, Steven L. Salzberg, Barbara J. Wold, and Lior Pachter. 2010. "Transcript Assembly and Quantification by RNA-Seq Reveals Unannotated Transcripts and Isoform Switching during Cell Differentiation." *Nature Biotechnology* 28 (5): 511–15.
- Wang, Dongxue, Basak Eraslan, Thomas Wieland, Björn Hallström, Thomas Hopf, Daniel Paul Zolg, Jana Zecha, et al. 2019. "A Deep Proteome and Transcriptome Abundance Atlas of 29 Healthy Human Tissues." *Molecular Systems Biology* 15 (2): e8503.
- White, Richard J., John E. Collins, Ian M. Sealy, Neha Wali, Christopher M. Dooley, Zsafia Digby, Derek L. Stemple, et al. 2017. "A High-Resolution mRNA Expression Time Course of Embryonic Development in Zebrafish." *eLife* 6 (November). <https://doi.org/10.7554/eLife.30860>.

February 1, 2022

Re: Life Science Alliance manuscript #LSA-2022-01397-T

Dr Alessandro Ori
Leibniz Institute on Aging - Fritz Lipmann Institute (FLI)
Beutenbergstrasse 11
Jena, Thuringia 7745
Germany

Dear Dr. Ori,

Thank you for submitting your manuscript entitled "Conserved exchange of paralog proteins during neuronal differentiation" to Life Science Alliance. We invite you to submit a revised manuscript addressing Reviewer 2's remaining concerns #2 and 3. This can be done via Discussion and by toning down some of the broader conclusions.

Thank you for this interesting contribution to Life Science Alliance. We are looking forward to receiving your revised manuscript.

Sincerely,

B. MANUSCRIPT ORGANIZATION AND FORMATTING:

Conserved exchange of paralog proteins during neuronal differentiation

We would like to thank the reviewers for the important criticism raised on our manuscript. Below we address point-by-point the main criticisms raised.

Regarding the important criticisms raised by Reviewer #2:

#1: Regarding the fact that some of the comments were not made on the first version. The major issue of multiple pairwise comparisons came up with the new analyses on the molecular divergence (Ka/Ks) of paralogs they added with the revision. It could not have been picked up in the first round.

#2: The issue with the Ka/Ks analysis is that values for very distant sequences are unreliable. And if the problem increases with sequence divergence, the estimates are more unreliable for more distant pairs. Any association between Ka/Ks ratio and some feature could be created by the association between the age of the paralogs and this feature. Also, it is usually rare to have Kn/Ks ratios above one and having many as they have here suggests that this saturation is an important issue. The issue is not that they do not acknowledge this effect, it is that this effect could be an artifact which they do not try to circumvent. The rebuttal here does not say how they want to improve this part.

This issue has been discussed extensively. Here is one example.

[https://www-cell-com.acces.bibl.ulaval.ca/trends/genetics/comments/S0168-9525\(02\)02722-1](https://www-cell-com.acces.bibl.ulaval.ca/trends/genetics/comments/S0168-9525(02)02722-1)

Here is what Ziheng Zhang writes about this issue in some forums on this issue.

A common problem in comparative analysis of genomes to estimate dS and dN is that the time scale or sequence divergence may be inappropriate. Estimation of dS and dN requires a time window in which the sequences are neither too similar nor too divergent. If the species are too distantly related or the genomes are too divergent, the synonymous substitutions may have reached saturation, so that it is impossible to obtain reliable estimates of dS. While any criterion is arbitrary, it appears prudent to treat estimates of dS greater than 3 with caution. It is virtually impossible to distinguish data with on average five changes per site from data with 50 changes per site, even though as estimates of dS, those values are very different. When the sequences are too divergent, one useful strategy may be to include other genomes and compare multiple species on a phylogenetic tree, thus breaking the large distance between species into many shorter branches. Such methods are discussed in Chapters 4 and 11.

We thank the reviewer for the important criticisms raised on our analysis. Here we want to clarify that the aim of our work was not to characterize the function of duplicated genes from an evolutionary point of view. The major focus of our manuscript is to characterize the role that paralog genes have in fine tuning molecular function in different cell types with a clear focus on macromolecular protein complexes, as exemplified by the COPII complex and Sec23 paralogs during the process of neuronal differentiation. Since the observation derived from the analysis on evolutionary divergence of paralog genes did not have a major impact on the broader conclusions of our work, we have decided to remove the corresponding section from our revised manuscript following the criticism of this reviewer and the independent expert below .

#3: It is true that keeping these reads or not keeping these reads can create problems in each case, but different kinds of problems. It is their role to convince us that their conclusions hold irrespective of the method they use, knowing the biases in each case.

My initial comment on this issue on the original version was the following: "It is not clear if in the data

analyses the RNAseq data was analysed in a way that takes into account cross mapping of reads between paralogs? If not, this could influence the correlation of expression across samples. If yes, this may lead to the elimination of many reads, which would increase the noise in the expression measurements and thus increase variation among samples, explaining some of the observations in Supp Fig1.

As you can see, I was not advocating one way or another, I was saying that the information was not provided. In either case, it can create biases in the analyses. The authors now mentioned how it was analyzed in the new version but did not address how it could affect their conclusion.

We thank the reviewer for specifying more in detail his concerns about the problem of estimating paralog abundances in the RNA-Seq datasets. It is indeed true that assigning reads that map to multiple, similar genes, can be challenging and lead to over- or under-estimation of correlation coefficients between paralog pairs. In the analysis of human tissue data, reads that mapped multiple times in the genome were equally split between genes. This might lead to an under-estimation of the percentage of divergent and anti-correlated paralog pairs (since it leads to a dilution of the signal). Thus, any signal of paralog expression divergence that we detected might represent a parsimonious estimate. Importantly, we validated the correctness of the transcriptome estimates by comparing them to proteome data that are based exclusively on proteotypic peptides that are by definition paralog specific. Thus, we are confident that the way RNA-Seq data were analyzed did not have a negative impact on the conclusion drawn in our manuscript.

We have included a new paragraph in the revised manuscript explaining the potential limitations of RNA-Seq data analysis:

Line 117-125, Page 3 : “The detection of divergently regulated paralogs in RNA-Seq data can be challenging due to the handling of reads that map to multiple paralog genes. Given how these reads were handled in the analyzed datasets, our estimates of divergent paralogs expression could be in some cases underpowered due to the equal splitting of shared reads between paralog genes (see Methods). Importantly, consistent results were obtained for human tissues using proteome data, which are based on proteotypic (unique) peptides that are by definition paralog-specific. The proportion of divergent paralogs estimated from the proteomics data appeared to be even higher (48%) than the estimates obtained from transcriptome data (Appendix Fig S1A).”

February 14, 2022

RE: Life Science Alliance Manuscript #LSA-2022-01397-TR

Dr. Alessandro Ori
Leibniz Institute on Aging - Fritz Lipmann Institute
Beutenbergstrasse 11
Jena, Thuringia 7745
Germany

Dear Dr. Ori,

Thank you for submitting your revised manuscript entitled "Conserved exchange of paralog proteins during neuronal differentiation". We would be happy to publish your paper in Life Science Alliance pending final revisions necessary to meet our formatting guidelines.

- please add a Running Title to our system
- please consult our manuscript preparation guidelines <https://www.life-science-alliance.org/manuscript-prep> and make sure your manuscript sections are in the correct order
- please add your main, supplementary figure, and table legends to the main manuscript text after the references section
- please remove the Panel A label from the legend for Figure S4

A. FINAL FILES:

B. MANUSCRIPT ORGANIZATION AND FORMATTING:

Sincerely,

Conserved exchange of paralog proteins during neuronal differentiation

We would like to thank the reviewers for the important criticism raised on our manuscript. Below we address point-by-point the main criticisms raised.

Regarding the important criticisms raised by Reviewer #2:

#1: Regarding the fact that some of the comments were not made on the first version. The major issue of multiple pairwise comparisons came up with the new analyses on the molecular divergence (Ka/Ks) of paralogs they added with the revision. It could not have been picked up in the first round.

#2: The issue with the Ka/Ks analysis is that values for very distant sequences are unreliable. And if the problem increases with sequence divergence, the estimates are more unreliable for more distant pairs. Any association between Ka/Ks ratio and some feature could be created by the association between the age of the paralogs and this feature. Also, it is usually rare to have Kn/Ks ratios above one and having many as they have here suggests that this saturation is an important issue. The issue is not that they do not acknowledge this effect, it is that this effect could be an artifact which they do not try to circumvent. The rebuttal here does not say how they want to improve this part.

This issue has been discussed extensively. Here is one example.

[https://www-cell-com.acces.bibl.ulaval.ca/trends/genetics/comments/S0168-9525\(02\)02722-1](https://www-cell-com.acces.bibl.ulaval.ca/trends/genetics/comments/S0168-9525(02)02722-1)

Here is what Ziheng Zhang writes about this issue in some forums on this issue.

A common problem in comparative analysis of genomes to estimate dS and dN is that the time scale or sequence divergence may be inappropriate. Estimation of dS and dN requires a time window in which the sequences are neither too similar nor too divergent. If the species are too distantly related or the genomes are too divergent, the synonymous substitutions may have reached saturation, so that it is impossible to obtain reliable estimates of dS. While any criterion is arbitrary, it appears prudent to treat estimates of dS greater than 3 with caution. It is virtually impossible to distinguish data with on average five changes per site from data with 50 changes per site, even though as estimates of dS, those values are very different. When the sequences are too divergent, one useful strategy may be to include other genomes and compare multiple species on a phylogenetic tree, thus breaking the large distance between species into many shorter branches. Such methods are discussed in Chapters 4 and 11.

We thank the reviewer for the important criticisms raised on our analysis. Here we want to clarify that the aim of our work was not to characterize the function of duplicated genes from an evolutionary point of view. The major focus of our manuscript is to characterize the role that paralog genes have in fine tuning molecular function in different cell types with a clear focus on macromolecular protein complexes, as exemplified by the COPII complex and Sec23 paralogs during the process of neuronal differentiation. Since the observation derived from the analysis on evolutionary divergence of paralog genes did not have a major impact on the broader conclusions of our work, we have decided to remove the corresponding section from our revised manuscript following the criticism of this reviewer and the independent expert below .

#3: It is true that keeping these reads or not keeping these reads can create problems in each case, but different kinds of problems. It is their role to convince us that their conclusions hold irrespective of the method they use, knowing the biases in each case.

My initial comment on this issue on the original version was the following: "It is not clear if in the data

analyses the RNAseq data was analysed in a way that takes into account cross mapping of reads between paralogs? If not, this could influence the correlation of expression across samples. If yes, this may lead to the elimination of many reads, which would increase the noise in the expression measurements and thus increase variation among samples, explaining some of the observations in Supp Fig1.

As you can see, I was not advocating one way or another, I was saying that the information was not provided. In either case, it can create biases in the analyses. The authors now mentioned how it was analyzed in the new version but did not address how it could affect their conclusion.

We thank the reviewer for specifying more in detail his concerns about the problem of estimating paralog abundances in the RNA-Seq datasets. It is indeed true that assigning reads that map to multiple, similar genes, can be challenging and lead to over- or under-estimation of correlation coefficients between paralog pairs. In the analysis of human tissue data, reads that mapped multiple times in the genome were equally split between genes. This might lead to an under-estimation of the percentage of divergent and anti-correlated paralog pairs (since it leads to a dilution of the signal). Thus, any signal of paralog expression divergence that we detected might represent a parsimonious estimate. Importantly, we validated the correctness of the transcriptome estimates by comparing them to proteome data that are based exclusively on proteotypic peptides that are by definition paralog specific. Thus, we are confident that the way RNA-Seq data were analyzed did not have a negative impact on the conclusion drawn in our manuscript.

We have included a new paragraph in the revised manuscript explaining the potential limitations of RNA-Seq data analysis:

Line 117-125, Page 3 : *“The detection of divergently regulated paralogs in RNA-Seq data can be challenging due to the handling of reads that map to multiple paralog genes. Given how these reads were handled in the analyzed datasets, our estimates of divergent paralogs expression could be in some cases underpowered due to the equal splitting of shared reads between paralog genes (see Methods). Importantly, consistent results were obtained for human tissues using proteome data, which are based on proteotypic (unique) peptides that are by definition paralog-specific. The proportion of divergent paralogs estimated from the proteomics data appeared to be even higher (48%) than the estimates obtained from transcriptome data (Appendix Fig S1A).”*

February 17, 2022

RE: Life Science Alliance Manuscript #LSA-2022-01397-TRR

Dr. Alessandro Ori
Leibniz Institute on Aging - Fritz Lipmann Institute
Beutenbergstrasse 11
Jena, Thuringia 7745
Germany

Dear Dr. Ori,

Thank you for submitting your Research Article entitled "Conserved exchange of paralog proteins during neuronal differentiation". It is a pleasure to let you know that your manuscript is now accepted for publication in Life Science Alliance. Congratulations on this interesting work.

DISTRIBUTION OF MATERIALS:

Again, congratulations on a very nice paper. I hope you found the review process to be constructive and are pleased with how the manuscript was handled editorially. We look forward to future exciting submissions from your lab.

Sincerely,
